# Neural control of growth and size in the axolotl limb regenerate

**Kaylee M Wells, Kristina Kelley, Mary Baumel, Warren A Vieira, Catherine D McCusker***

Biology Department, University of Massachusetts, Boston, Boston, United States

**Abstract** The mechanisms that regulate growth and size of the regenerating limb in tetrapods such as the Mexican axolotl are unknown. Upon the completion of the developmental stages of regeneration, when the regenerative organ known as the blastema completes patterning and differentiation, the limb regenerate is proportionally small in size. It then undergoes a phase of regeneration that we have called the 'tiny-limb' stage, which is defined by rapid growth until the regenerate reaches the proportionally appropriate size. In the current study we have characterized this growth and have found that signaling from the limb nerves is required for its maintenance. Using the regenerative assay known as the accessory limb model (ALM), we have found that growth and size of the limb positively correlates with nerve abundance. We have additionally developed a new regenerative assay called the neural modified-ALM (NM-ALM), which decouples the source of the nerves from the regenerating host environment. Using the NM-ALM we discovered that non-neural extrinsic factors from differently sized host animals do not play a prominent role in determining the size of the regenerating limb. We have also discovered that the regulation of limb size is not autonomously regulated by the limb nerves. Together, these observations show that the limb nerves provide essential cues to regulate ontogenetic allometric growth and the final size of the regenerating limb.

**\*For correspondence:**
catherine.mccusker@umb.edu

**Competing interest:** The authors declare that no competing interests exist.

## Editor's evaluation

It has long been known that nerves regulate the early formation of the blastema during limb regeneration through the promotion of cell proliferation. The manuscript provides an interesting new role for nerves during salamander limb regeneration by showing that nerves also determine how much tissue to regenerate. They demonstrate that increased nerve abundance makes bigger limbs while a decrease in nerve abundance generates smaller limbs. Size regulation of organs is a broadly interesting and clinically important problem, which is why this manuscript should be of interest to a large general audience.

## Introduction

It is estimated that over 2.1 million Americans are living with limb loss or limb difference, which has profound effects on the function, health, and quality of life in these patients (*Ziegler-Graham et al., 2008*). The long-term goal of regenerative medicine is to replace or repair damaged limbs by inducing endogenous regenerative responses in humans. Studies in tetrapods capable of regenerating complete and functional limb structures, such as the Mexican axolotl (*Ambystoma mexicanum*), have been invaluable in terms of understanding the basic underlying biology of limb regeneration and the mechanisms that control this process (*McCusker et al., 2015*). Much research in the axolotl system has focused on the essential aspects of the initial stages of regeneration; how the regeneration permissive environment is established, how mature limb cells become regeneration competent, and how the unique pattern of the regenerated limb structures is generated. However, to date, very little

**eLife digest** Humans' ability to regrow lost or damaged body parts is relatively limited, but some animals, such as the axolotl (a Mexican salamander), can regenerate complex body parts, like legs, many times over their lives. Studying regeneration in these animals could help researchers enhance humans' abilities to heal. One way to do this is using the Accessory Limb Model (ALM), where scientists wound an axolotl's leg, and study the additional leg that grows from the wound.

The first stage of limb regeneration creates a new leg that has the right structure and shape. The new leg is very small so the next phase involves growing the leg until its size matches the rest of the animal. This phase must be controlled so that the limb stops growing when it reaches the right size, but how this regulation works is unclear. Previous research suggests that the number of nerves in the new leg could be important.

Wells et al. used a ALM to study how the size of regenerating limbs is controlled. They found that changing the number of nerves connected to the new leg altered its size, with more nerves leading to a larger leg. Next, Wells et al. created a system that used transplanted nerve bundles of different sizes to grow new legs in different sized axolotls. This showed that the size of the resulting leg is controlled by the number of nerves connecting it to the CNS. Wells et al. also showed that nerves can only control regeneration if they remain connected to the central nervous system.

These results explain how size is controlled during limb regeneration in axolotls, highlighting the fact that regrowth is directly controlled by the number of nerves connected to a regenerating leg. Much more work is needed to reveal the details of this process and the signals nerves use to control growth. It will also be important to determine whether this control system is exclusive to axolotls, or whether other animals also use it.

is known about the later stages of regeneration that are required for the regenerating structure to mature into a fully functional limb.

One key occurrence at the later stages of limb regeneration is the growth of the limb regenerate to the size that is proportionally appropriate to the rest of the animal. When the limb is amputated, a transient regenerative organ known as the blastema develops at the site of injury and contains limb progenitor cells that will grow and eventually pattern and differentiate into the missing limb structure. After these developmental (blastema) stages of regeneration, the limb is small in size relative to the animal's body and undergoes a period of rapid ontogenetic allometric growth until the size of the regenerate is proportional to the rest of the animal. Ontogenetic allometric growth refers to the rate of growth of a structure (i.e. the limbs) relative to the rest of the animal. Little is known about how this growth is regulated to impact the overall scaling of the regenerated limb. Moreover, axolotl are an indeterminately growing species, and continue to grow in size throughout their life cycle. Thus, the size of the limb at the time of amputation is different from that once the limb has completed regeneration (*Riquelme-Guzmán et al., 2021*). This simple observation indicates that rather than having a 'set-point' of size, growth must be dynamically regulated throughout the process of limb regeneration in the axolotl.

Most of what is known about the factors that impact tetrapod limb size has been discovered by mutant analyses on determinately growing species, such as humans, mice, and chicken, which cease growing after they reach adulthood due to the closure of the epiphyseal growth plates. These studies reveal that mutations to genes that play roles in patterning, long bone growth, or overall cell physiology in the developing limb tissues can have large impacts on the final size of the limb. For example, the developing bat forelimb bud has increased expression of HoxD transcription factors compared to hindlimb bud and is required for the extreme positive allometric growth that occurs in this tissue as it grows into a wing (*Booker et al., 2016*). At the later stages of tetrapod limb formation, allometric growth is primarily driven through the elongation of the long bones through the regulation of growth plate activity. As a result, alterations in expression of genes that impact paracrine signals which regulate the growth plates, such as fibroblast growth factor (*FGF*), bone morphogenetic protein (*BMP*), and Indian hedgehog (*IHH*), can all positively and negatively regulate the final size of the limb (*Iwata et al., 2001*; *Iwata et al., 2000*; *Lee et al., 2017*; *Segev et al., 2000*; *Eswarakumar and Schlessinger, 2007*; *Toydemir et al., 2006*; *Tseng et al., 2010*;

*Wen et al., 2016*; *Duprez et al., 1996*; *Aizawa et al., 2012*; *Barna et al., 2000*; *Bhattacharyya et al., 2015*; *Evers et al., 1996*; *Karst et al., 1986*; *Klüppel et al., 2005*; *Lallemand et al., 2005*; *Zhang et al., 2020*; *Caparrós-Martín et al., 2013*; *Joeng and Long, 2009*; *Kim et al., 2017*; *Long et al., 2001*; *Mo et al., 1997*; *Razzaque et al., 2005*; *Ruiz-Perez et al., 2007*; *Sohaskey et al., 2008*; *St-Jacques et al., 1999*; *Yoshida et al., 2015*; *Zhang et al., 2015*; *Bren-Mattison et al., 2011*; *Minina et al., 2002*). Thus, ontogenetic allometric growth in the developing limb bud can be impacted by intrinsic or extrinsic factors at both the developmental and maturation stages of limb formation.

Given the similarities between the developing limb bud and the regenerating limb blastema, these mechanisms are likely to hold true during limb regeneration. Supporting this notion, studies using pharmacological inhibitors for FGF, BMP, or TGFβ signaling on the developing axolotl limb blastema all result in smaller limbs compared to controls by negatively impacting patterning, differentiation, or overall blastema physiology (*Lévesque et al., 2007*; *Purushothaman et al., 2019*; *Vincent et al., 2020*). Conversely, treatment of the axolotl blastemas with exogenous retinoic acid (RA), which negatively regulates *HoxA13* (a marker of distal limb) (*Gardiner et al., 1995*) during blastema patterning, results in the elongation of skeletal elements in the regenerated limb (*Maden, 1983*; *Niazi et al., 1985*). However, how these and other signals that regulate limb growth are so tightly coordinated during regeneration such that the size of the completed limb is proportional to the animal is not known. Identifying the source of this coordination is the focus of the current study.

Multiple pieces of evidence point to the possible involvement of neural signaling in the regulation of allometric growth in limbs. In amphibians, neural signaling is an essential component of the developmental stages of limb regeneration in both the induction and maintenance of the blastema (*Singer, 1978*). Nerve signaling is essential for proliferation in the blastema, and loss of innervation at early stages of regeneration results in complete regenerative failure (*Singer, 1978*). Multiple protein-based nerve-dependent factors have been identified to play key roles in the maintenance of the limb blastema including anterior gradient (AG), BMPs, FGFs, and neuregulin-1 (*Kumar et al., 2007*; *Makanae et al., 2014*; *Farkas et al., 2016*). At the late-bud blastema stage, when the pattern of the regenerated limb has been established, the loss of nerve signaling results in the formation of complete, yet miniaturized, limb regenerates (*Mullen et al., 1996*). Last, permanently miniaturized limbs, generated by the repeated removal of the limb bud, have a significant decrease in the abundance of innervation (*Bryant et al., 2017*). Thus, the loss of neural signaling during the developmental stages of limb formation in the axolotl correlates with negative ontogenetic allometric growth and smaller limb size.

Nerve abundance, or altered nerve signaling, also correlates with growth and limb size in humans. Damage to the limb nerves during the birth of a human infant results in impaired limb growth and size, which can be alleviated by surgical repair of the nerves (*Bain et al., 2012*). Additionally, the presence of neurofibromas or hamartomas in human arms or hands, which increases the abundance of neural signals, results in the formation of proportionally large digits (*Frykman and Wood, 1978*; *Razzaghi and Anastakis, 2005*; *Tsuge and Ikuta, 1973*). These observations support the idea that neural signaling plays a role in the regulation of growth in the tetrapod limbs. In the current study, we directly test this possibility in the regenerating axolotl limb.

Here, we characterize the growth in the regenerating limb and reveal that its growth is biphasic. By removing nerve signaling from regenerating limbs at different stages, we show that neural signaling is required for multiple cellular behaviors that contribute to growth, and that this requirement changes during the different phases of growth. Using the accessory limb model (ALM) assay, our data shows that nerve abundance positively correlates with the rate of growth and the ultimate size of the limb regenerates. To evaluate the potential role of non-neural signals in the regulation of growth, we have developed a new assay that we call the NM-ALM, which decouples the nerve source from the host animal. Using this assay, we found that non-neural extrinsic factors do not play a direct instructive role in the regulation of scaling in the regenerating limb tissue. We have additionally used the NM-ALM to evaluate whether neurons work autonomously to regulate growth in the regenerate and have discovered that they require a connection with their native environment to coordinate this process. These observations will be foundational to future work on the identification of the molecular mechanisms that regulate ontogenetic allomeric growth of the regenerating axolotl limb.

## Results

### The axolotl limb undergoes at least three stages of growth during regeneration

Because there was so little known about how allometric growth occurs in the regenerating limb, we started by characterizing the growth of the regenerate until it had completed regeneration. We measured limb and body length to calculate both limb 'proportionality' (ratio of limb length to body length) (*Figure 1—figure supplement 1*) and the growth rate in 10 cm sized animals over a period of 140 days, which is when the regenerated limb was no longer significantly different in size relative to the uninjured limbs on the same sized animal, and the growth rate of these limbs was also equivalent. Because axolotls are indeterminate growers, both their limb and body lengths continue to increase in size during the course of regeneration, thus the ratio of limb length to body length was used to normalize for the changes in animal size as time proceeded.

We observed that the regenerating limb underwent at least three phases of growth before it reached the size of the unamputated limb, which occur during the blastema and the post-developmental stage of regeneration (*Figure 1A and B*). The growth during the blastema stage starts immediately following amputation all the way through the digit stage of blastema development. Previous studies have shown that the expression of growth factors and signaling molecules associated with blastema development are lost by the digit staged regenerate, thus indicating the end of the blastema stage, and initiation of post-developmental regenerative processes (day 38) (*Gerber et al., 2018*; *Nacu et al., 2016*; *Satoh et al., 2008*). The digit staged regenerate is significantly smaller than the unamputated limb on size-matched animals (*Figure 1A and B*). This small regenerate undergoes rapid ontogenetic allometric growth until it has reached the proportionally appropriate size, at which point the regenerate grows isometrically with the rest of the animal. We have named this post-blastema staged regenerate the 'tiny limb'. We observed that the allometric growth is biphasic in the tiny limb. The initial phase of growth is rapid, similar to the speed of growth during the blastema stage at 0.04 cm/day (*Figure 1B* and Figure S2A). Approximately 4 weeks later (in 10 cm sized animals), the rate of growth of the tiny limb slows significantly to 0.02 cm/day over the following 9 weeks and gradually decreases over time, until both the size and the growth rate of the regenerated limb are not significantly different than the unamputated limb on size-matched animals (*Figure 1B* and *Figure 1—figure supplement 2A*). Based on this difference in the rate of allometric growth, we have separated the tiny limb stage of development into two phases of growth: the early tiny limb (ETL) stage and the late tiny limb (LTL) stage.

Both the growth rate and the amount of time spent in each of the three stages (blastema, ETL, and LTL) are dependent on the size of the animal at the time of limb amputation. Comparison between 4, 10, and 20 cm animals (snout to tail tip) showed that as body length increases, the growth rate decreases (*Figure 1—figure supplement 2*), and the amount of time spent in the ETL and LTL phase increases (*Figure 1C*). In fact, when comparing animals twofold different in size, the amount of time in the ETL phase increases by 30–40% (*Figure 1C*) and growth rate falls by 30–40% (*Figure 1—figure supplement 2*). This difference in growth rate (measured in terms of regenerate elongation) may be due to the difference in the amount of tissue that needs to be regenerated, since larger animals have more tissue to regenerate than the smaller animals. Additionally, smaller animals grow more rapidly than larger animals, which also might contribute to these differences (*Figure 1—figure supplement 2B*).

### Growth of the tiny limb is mediated by increased cell number and cell size

We next wanted to determine what cellular mechanisms were contributing to growth in the tiny limb. Multiple processes can contribute to tissue growth including increased cell number via regulation of cell proliferation and cell death, increased cell size, and increased extracellular matrix (ECM) deposition (*Conlon and Raff, 1999*; *Leevers et al., 1996*; *Penzo-Méndez and Stanger, 2015*; *Stanger, 2008*; *Stocker and Hafen, 2000*). Thus, we quantified each of these processes during the different growth phases of regeneration, relative to uninjured limbs, to determine which could be contributing to growth of the tiny limb. Additionally, we speculated that the contribution of these cell processes could vary in the different tissue types in the regenerating limb. Rather than quantifying

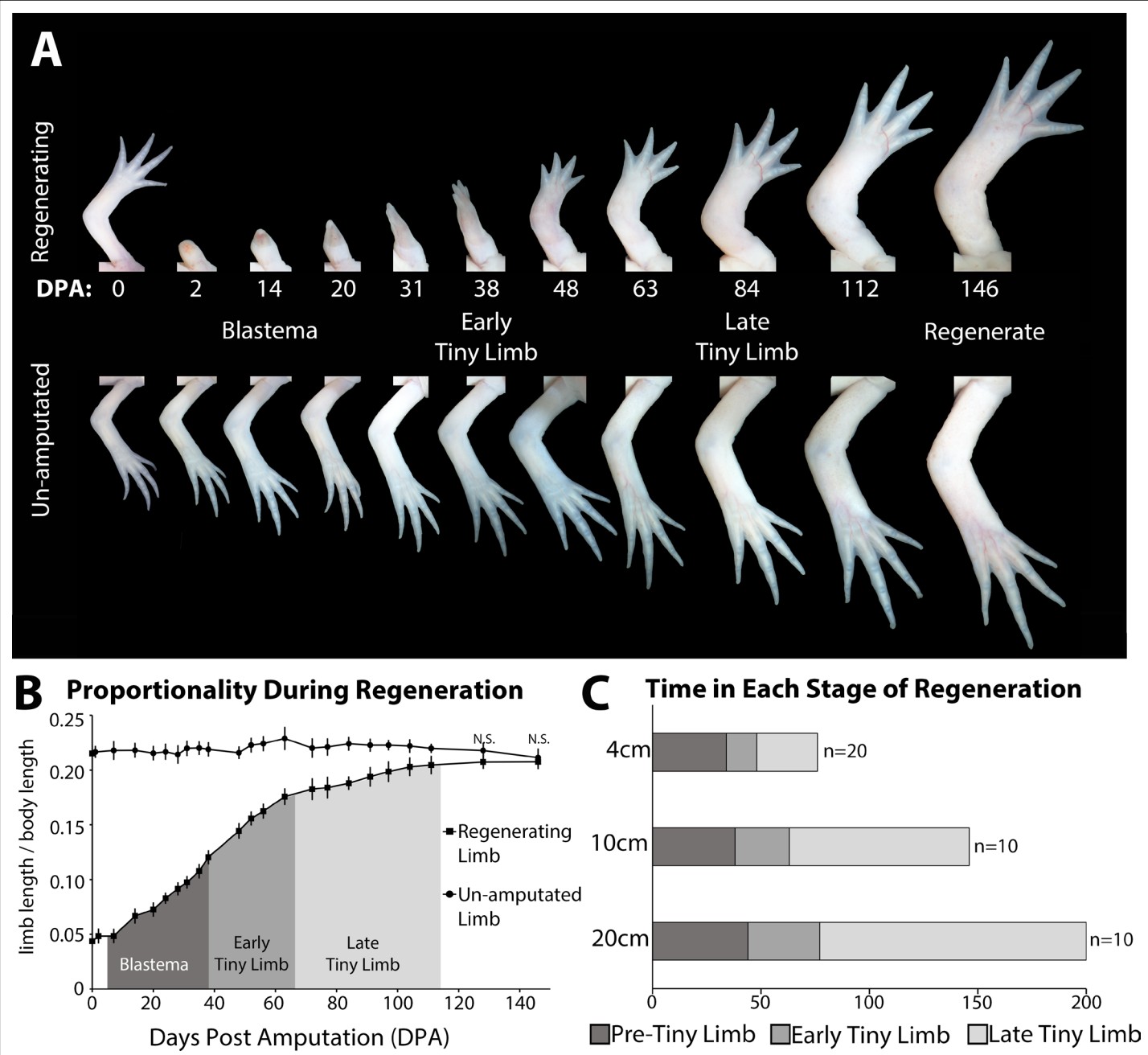

**Figure 1.** The tiny limb grows at an increased rate compared to an unamputated limb. (**A**) Time course of growth in amputated (top panel) and the contralateral non-amputated (lower panel) limbs on a 10 cm animal over 146 days. (**B**) The ratio of limb to body length in regenerating and unamputated limbs was measured over time (10 cm animals; n = 10). We have separated the growth of the limb regenerate into three stages: the blastema stage (dark gray), the early tiny limb stage (medium gray), and the late tiny limb stage (light gray). Error bars = standard deviation. t-Test was used to evaluate significance between the regenerating and uninjured limb size at each time point. All data points not marked with N.S. had p-values less than 0.005. (**C**) Histogram showing the average amount of time in days that the regenerating limb is in each growth stage for animals of different sizes (4, 10, and 20 cm in length).

The online version of this article includes the following figure supplement(s) for figure 1:

**Figure supplement 1.** Axolotl size measurements.

**Figure supplement 2.** Animal size corresponds with growth rate during limb regeneration.

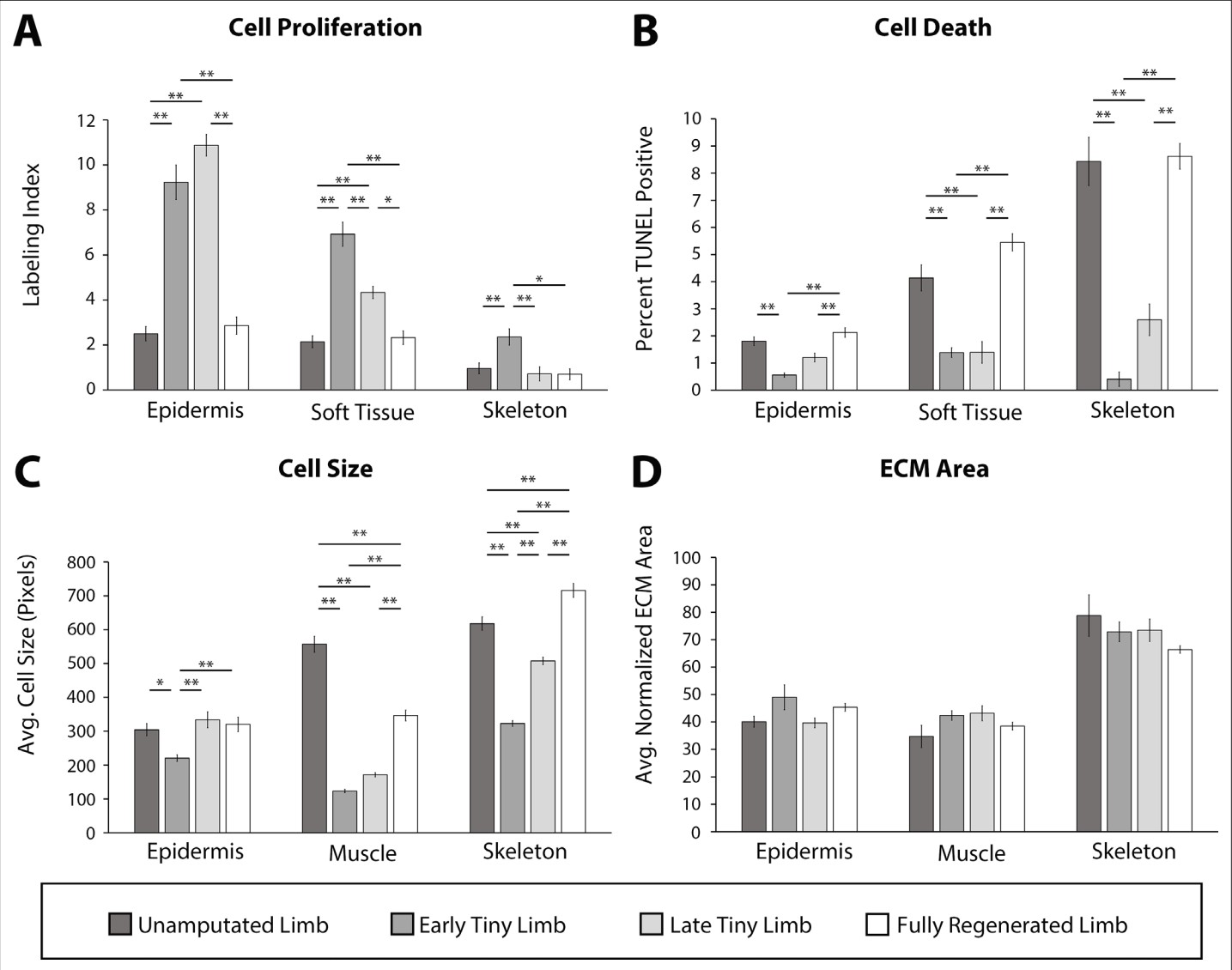

**Figure 2.** Tiny limb staged regenerates have increased proliferation, decreased cell death, and smaller cell sizes than uninjured or completely regenerated limbs. Transverse sections through the zeugopod of limbs at different stages of regeneration were analyzed for cell proliferation (**A**), cell death (**B**), cell size (**C**), and extracellular matrix (ECM) size (**D**). (**A and B**) Cell proliferation and death were analyzed in the epidermis, soft tissue, and skeletal elements. (**A**) Cell proliferation was analyzed by EdU labeling (n = 5). (**B**) Cell death was analyzed using TUNEL labeling (n = 4). (**C and D**) Cell and ECM size measurements were quantified in the epidermis, muscle, and skeletal elements. (**C**) Cell size was quantified using fluorescently tagged wheat germ agglutinin (plasma membrane) for epidermal and muscular analysis and Alcian blue staining (collagen) for skeletal analysis (n = 4). (**D**) ECM area was calculated by [(tissue area – cellular area)/tissue area] (n = 4). Error bars = SEM. p-Values calculated by ANOVA and the Tukey post hoc test. * = p < 0.05 ** = p < 0.005.

The online version of this article includes the following figure supplement(s) for figure 2:

**Figure supplement 1.** Measurement of cell and extracellular matrix (ECM) size.

the above-described processes globally, we analyzed the epidermis, soft tissue (including all tissues except for skeleton and epidermis), and skeletal tissue (bone and cartilage) separately.

Regenerating limbs at the different stages of growth were sectioned transversely mid-zeugopod, and cell proliferation and cell death were each analyzed by either EdU incorporation or TUNEL staining, respectively. We observed significantly more cell proliferation and significantly less cell death in all tissues analyzed in the ETL and LTL staged regenerates compared to the unamputated and fully regenerated limbs (*Figure 2A and B*). Interestingly, the fold increase or decrease in cell proliferation or death, respectively, differed depending on the tissue type. The largest increase in cell

proliferation was observed in the epidermis (3.7-fold increase, *Figure 2A*), while soft tissue and skeletal tissue had slightly more modest increases (3.2- and 2.4-fold increases respectively, *Figure 2A*). The largest decrease in cell death was seen in the skeletal tissue (20.8-fold decrease, *Figure 2B*), while the soft tissues and epidermis exhibited more moderate decreases of 3.0- and 3.3-fold, respectively (*Figure 2B*).

Cell size and ECM area were analyzed using a combination of fluorescent and histological stains on sectioned limbs. Wheat germ agglutinin (WGA) was used to label the plasma membrane of the epidermal and muscle cells, and Alcian blue stained the ECM of the skeletal tissue (*Figure 2—figure supplement 1*) (more detail in Materials and methods). To standardize our quantification of the average cell size, we measured only the area of cells where the nucleus was observable (*Figure 2—figure supplement 1A*). We observed that cell size was significantly smaller in the regenerating tissue than the uninjured tissue and increased as regeneration progressed (*Figure 2C*). This was most profound in the muscle (4.5-fold smaller) and least in the epidermis (1.4-fold smaller, *Figure 2C*). The ECM area was calculated indirectly by subtracting the total cellular area from the tissue area and dividing by the tissue area (*Figure 2—figure supplement 1B*) (more detail in Materials and methods). However, we did not observe any significant differences in the extracellular compartment of limbs, indicating that ECM deposition does not play a significant role in growth of the tiny limb (*Figure 2D*).

We also observed stage-specific differences in the above-described growth characteristics. In almost all tissues, the largest differences were observed between the ETL stage and the unamputated limb, while the differences between the LTL and the unamputated limb were more modest. Apart from cell size in the muscle tissue, there were no significant differences in the growth characteristics when the unamputated and fully regenerated limb were compared.

Together, this data indicates that a combination of increased cell proliferation, decreased cell death, and increased cell size contributes to the allometric growth of the tiny limb staged regenerate. While all tissue types showed the same trends in all the cell processes that we analyzed, our data suggests that different cell processes contribute more or less to growth in different tissue types. Future studies will be required to resolve these tissue-specific contributions to growth in more detail. Last, the tissue-specific contributions to growth also depend on the stage of growth of the regenerate and correlate with the overall growth rate of the regenerate at these stages.

## Growth of the tiny limb is dependent on limb nerves

One interesting observation from the above-described characterization is that the abundance of cell proliferation, cell death, and cell size all show similar trends regardless of the tissue type assessed during each stage of growth in the regenerate. This suggests that there could be a singular signal

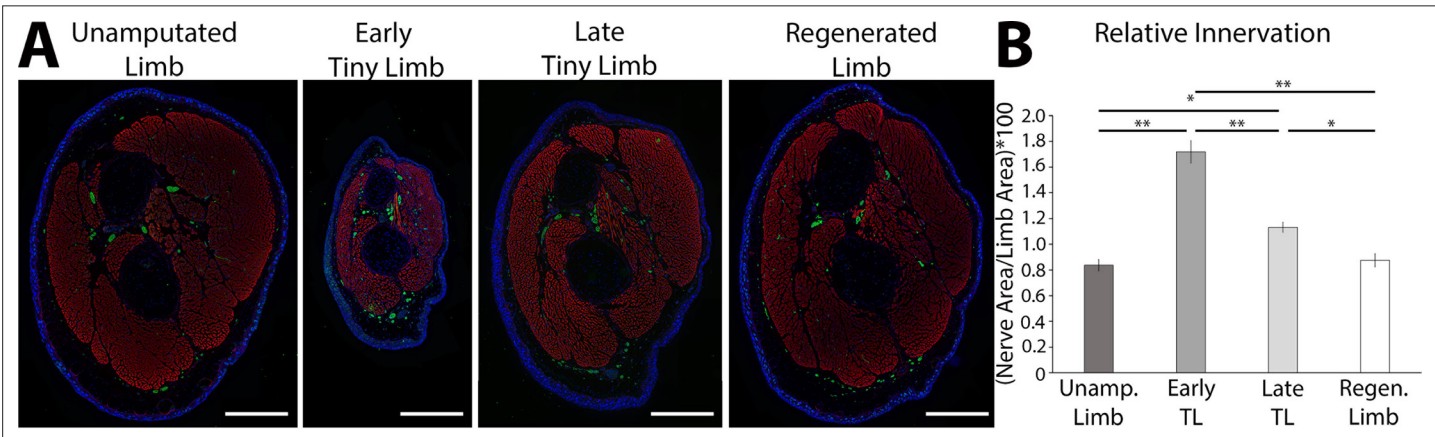

**Figure 3.** The tiny limb staged regenerate has increased relative nerve abundance. (**A**) Fluorescent images were obtained of transverse sections of uninjured, early, and late tiny limb stages, and fully regenerated limbs (DAPI = blue, phalloidin (for actin filaments) = red, acetylated tubulin (for nerves) = green; scale bars are 1000 μm). (**B**) Nerve area relative to total limb area was quantified from the sections represented in A (n = 5). Error bars = SEM. p-Values calculated by ANOVA and the Tukey post hoc test. * = p < 0.05 ** = p < 0.005.

The online version of this article includes the following figure supplement(s) for figure 3:

**Figure supplement 1.** Absolute abundance of nerves in different staged regenerates.

that coordinates these processes such that the highest growth-promoting signal is occurring during the ETL stage when growth is most abundant and decreases as the growth rate slows during the LTL stage. Thus, we next sought to determine the source of the signal that regulates cell proliferation, death, and size during regeneration.

Previous studies indicate that nerve signaling is required for growth in developing and regenerating limbs. The role and mechanism of neurotropic regulation during the early (blastema) stages of regeneration has been widely studied (*Farkas et al., 2016*; *Farkas and Monaghan, 2017*; *Kumar et al., 2010*; *Makanae et al., 2014*; *Singer, 1978*; *Singer, 1952*; *Singer, 1946*; *Singer and Inoue, 1964*). It has been well established that nerve signaling is required for, and is a key driver of, blastemal cell proliferation (*Brockes, 1984*; *Brockes and Kintner, 1986*; *Lehrberg and Gardiner, 2015*). Furthermore, generation of permanently miniaturized limbs through repeated removal of limb buds results in 'mini limbs' that are hypo-innervated compared to controls (*Bryant et al., 2017*). We therefore hypothesized that nerves could play a role in driving growth during the tiny limb stages of regeneration.

To test this idea, we first characterized the abundance of nerves during these post-developmental stages of regeneration. We collected ETL and LTL staged regenerates, as well as unamputated and fully regenerated limbs for comparison, and sectioned them transversally through the zeugopod. The sections were stained with an anti-acetylated tubulin antibody (*Figure 3A*), and we measured the abundance of innervation relative to the limb area (*Figure 3B*). This quantification revealed significantly higher levels of relative innervation during the ETL and LTL stages compared to unamputated and fully regenerated limbs (*Figure 3B*). Interestingly, the absolute abundance (not normalized to tissue area) of innervation is lowest at the ETL stage and increases during the later stages of regeneration . Thus, the decrease in relative innervation during the transition from the ETL to LTL stages is likely due to the substantial increase in limb area of the later staged regenerate. The relative abundance of innervation also correlates well with the growth rate in these tissues. We speculated that nerves could provide growth-promoting signals during regeneration, which decreases (or is diluted) as the limb grows in size and the relative abundance of innervation decreases, slowing the growth of the regenerate as it reaches its final size.

To determine whether nerve signaling plays a functional role regulating ontogenetic allometric growth, we next tested whether nerve signaling is required to maintain growth in the tiny limb staged regenerate. Limbs were amputated and permitted to regenerate to the ETL stage, at which point nerve signaling was severed via denervation at the brachial plexus (*Figure 4A*). To test for a possible dose-response or signaling threshold effect, we severed either one, two, or all three of the nerve bundles at the plexus. Mock denervation surgeries were performed as controls. The limbs were measured prior to denervation and 4 days post denervation, when they were collected for analysis. The growth rate, abundance of cell proliferation, abundance of cell death, and cell size were all analyzed (*Figure 4B–E*).

We observed that nerve signaling is required for growth of the tiny limb. Fully denervated ETLs had a 9-fold slower growth rate than ETLs with the nerve intact (*Figure 4B*). Likewise, cell proliferation was negatively impacted by denervation in all tissue types analyzed (1.7-, 4.9-, and 4.8-fold decreases in the epidermis, soft tissue, and skeleton, respectively; *Figure 4C*). Cell death levels in all tissues were significantly increased following full denervation (21.6-, 1.6-, and 25.7-fold increases in epidermis, soft tissue, and skeleton, respectively; *Figure 4D*). Lastly, cell size appears to only be significantly affected in the muscle, where there is a near 2-fold decrease in average cell size in the denervated ETLs (*Figure 4E*). It is possible that this change in muscle could be due to atrophy as a result of decreased muscle activity in the denervated limb. When LTLs were fully denervated, only the growth rate and abundance of cell proliferation were significantly decreased (*Figure 4—figure supplement 1*). Thus, the impact of neural signaling is different depending on whether the regenerating limb is in the early or late phase of growth.

The partial denervations revealed either a dose-response or threshold response depending on the tissue and growth characteristic quantified. A dose response is reflected by a linear relationship between abundance of signal and the phenotype. A threshold response indicates a specific abundance of a factor is required for a phenotype, for example, a specific amount of nerves per limb volume is required for growth. We observed that cell proliferation in the soft tissue and skeleton decreased significantly, to full denervation levels, with partial denervations indicating that there is a high threshold of nerve abundance required for maintaining cell proliferation in these tissues (*Figure 4C*). Conversely,

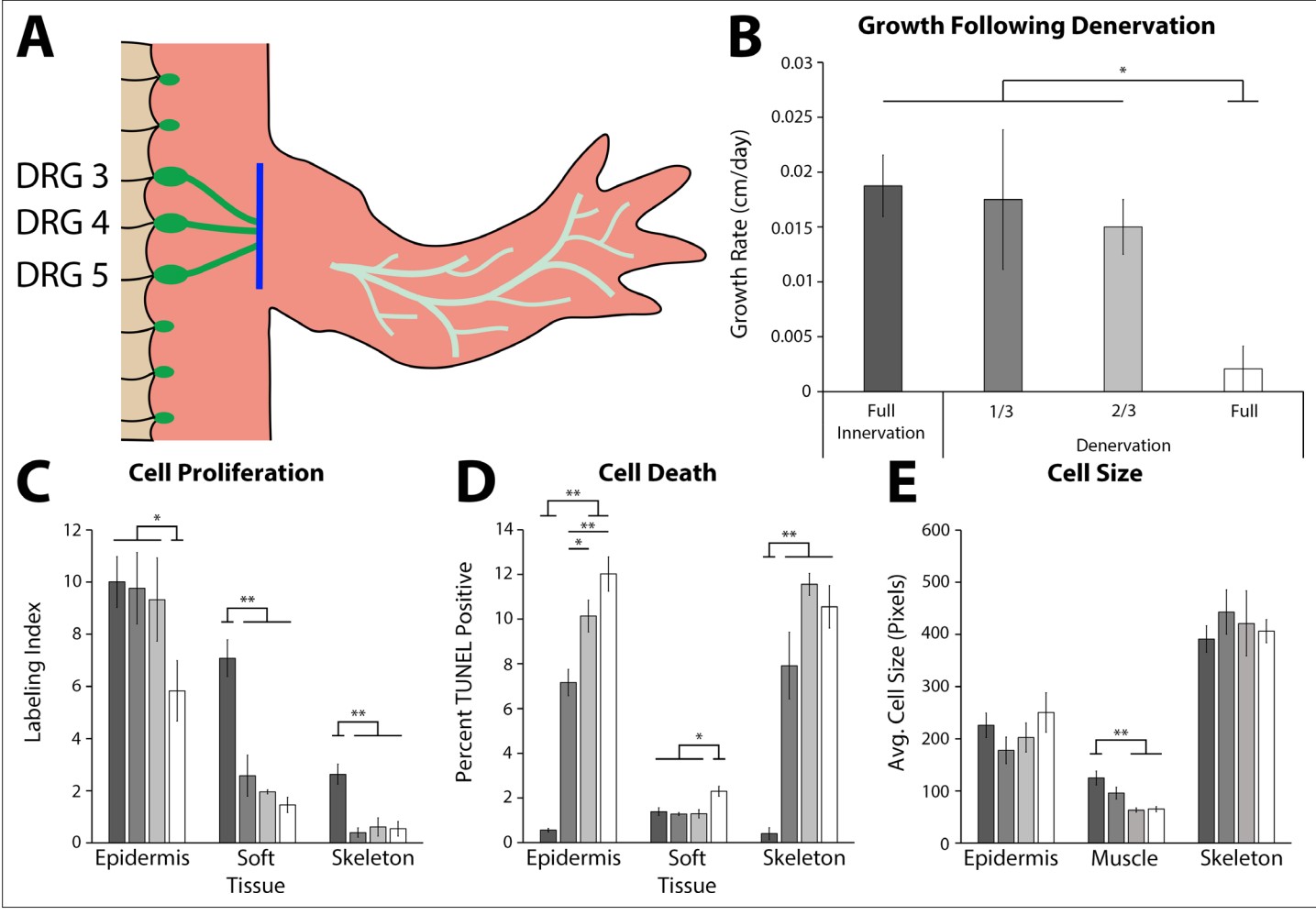

**Figure 4.** The effect of denervation on the growth of the tiny limb staged regenerate. (**A**) Dorsal root ganglia (DRGs) 3, 4, and 5 (green dots) are located lateral to the spinal column and their nerve bundles (green lines) feed into the forelimbs. Limbs were amputated and permitted to regenerate to the early tiny limb stage, at which point, either a mock, partial (1/3 = DRG 5 or 2/3 = DRGs 4 and 5), or full denervation (represented) was performed by severing (blue line) and removing sections of the nerve bundles. Limbs were collected 4 days post denervation, and growth rate (**B**), cell proliferation (**C**), cell death (**D**), and cell size (**E**) were analyzed for limbs with mock denervations (n = 6), 1/3 denervations (n = 5), 2/3 denervations (n = 5), and full denervations (n = 6). The color of the bars in panels C–E refers to the color of the bars in panel B. Error bars = SEM. p-Values calculated by ANOVA and the Tukey post hoc test. * = p < 0.05 ** = p < 0.005.

The online version of this article includes the following figure supplement(s) for figure 4:

**Figure supplement 1.** Impact of denervation on the growth characteristics of the late tiny limb (LTL) staged regenerate.

cell death in the epidermis had a strong dose response, with significant incremental increases with increased denervation (*Figure 4D*). These results indicate that each tissue responds differently to nerve abundance to maintain growth. This differential contribution of the different limb tissues could also explain why the overall growth rate of the regenerate is significantly impacted only when a full denervation is performed (*Figure 4B*).

## Nerve abundance correlates with growth rate and overall size of the regenerated limb

Having established that nerves are required for growth during the late stages of limb regeneration, we next wanted to determine whether we could positively and negatively manipulate the size of the regenerate by altering the abundance of nerves the regenerating tissue is exposed to. To test this, we performed a grafting experiment between differently sized axolotl (*Figure 5A*). The size of the nerve bundles originating from the spinal column increases as the animal grows, and quantification of

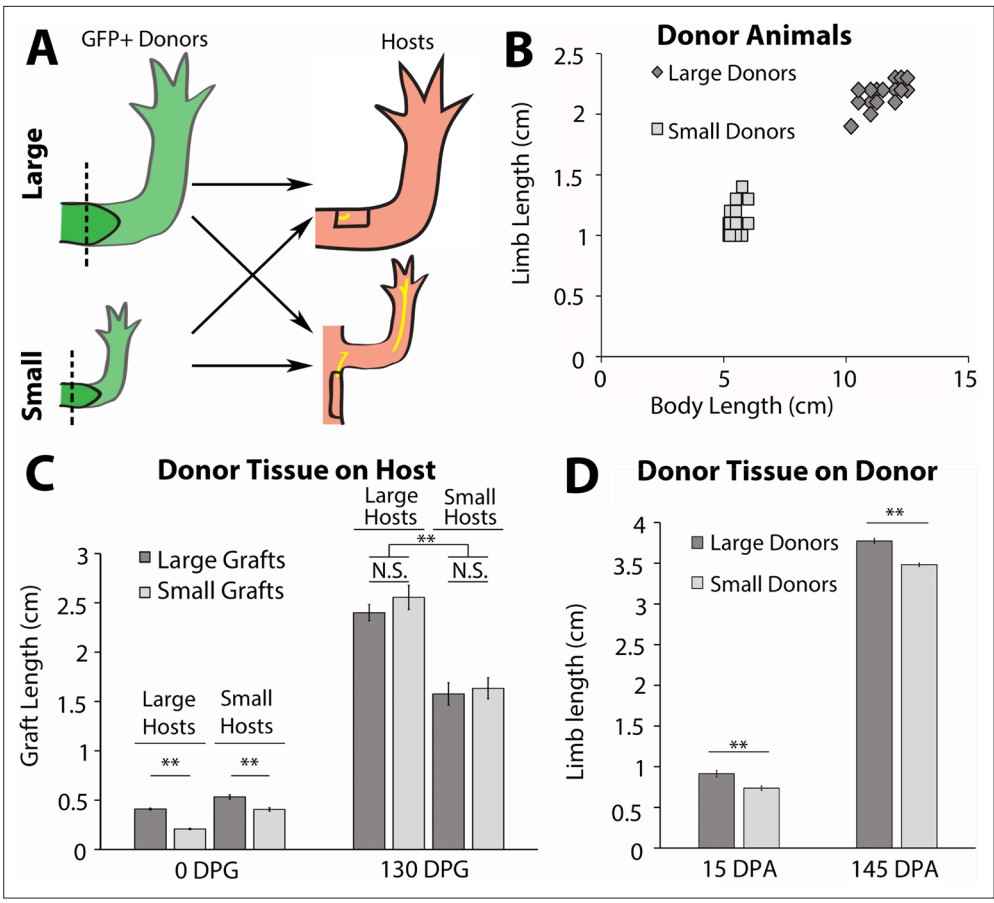

**Figure 5.** Host size correlates with the final size for the regenerated limb in the accessory limb model (ALM) assay. (**A**) Blastemas with approximately 2 mm of stump tissue from large and small GFP+ donor animals were grafted onto a regenerative permissive environment, a wound site with a deviated limb nerve bundle, on large or small host animals. (**B**) Limb length and body length were measured on the GFP+ donor animals. (**C**) The regenerating grafted tissues were measured at 0 and 130 days post graft (DPG). Blastemas from large donors (dark gray) were grafted onto large (n = 7) and small (n = 9) host animals, and blastemas from small donors (light gray) were grafted onto large (n = 9) and small (n = 10) host animals. (**D**) The regenerating large (n = 10) and small (n = 20) animal donor limbs were measured at 15- and 145 days post amputation. Error bars = SEM. p-Values calculated by ANOVA and the Tukey post hoc test. * = p < 0.05 ** = p < 0.005.

The online version of this article includes the following figure supplement(s) for figure 5:

**Figure supplement 1.** Size of nerve bundles in the peripheral nervous system correlate with animal size.

**Figure supplement 2.** Growth of grafted limbs in accessory limb models (ALMs) and neural-modified ALMs (NM-ALMs).

the cross-sectional area of these bundles stained immunofluorescently in situ reveals that the size is almost 2-fold larger in 14 cm long animals compared to 7 cm animals (***Figure 5—figure supplement 1***). Thus, blastema grafts onto the limbs of large hosts will be exposed to larger nerve bundles than grafts on small host animals.

To generate large and small animals we housed age-matched GFP+ (donors) and GFP- (hosts) axolotl at either 19°C or 4°C. Housing half the animals at 4°C stunted their growth while the 19°C animals grew at a faster rate. After approximately 2 months, the 19°C animals were 2-fold larger in body length and limb length than the 4°C animals (***Figure 5B***). After the large and small siblings were both incubated at 19°C for 14 days, both forelimbs on small (~6 cm) and large (~12 cm) GFP+ axolotl were amputated and permitted to regenerate to the mid-bud blastema stage (***Figure 5A and B***). The blastemas and approximately 2 mm of stump tissue were grafted onto regenerative permissive environments on small (~6 cm) and large (~12 cm) GFP-host animals (***Figure 5A***). It has previously been found that cells from approximately 500 µm of stump tissue migrate and contribute to the regenerate

(*Currie et al., 2016*). Therefore, stump tissue was included with the blastemas to prevent the contribution of cells from the host environment to the regenerate.

A regeneration permissive environment on the large animal hosts was generated by deviating the branchial nerve bundle to an anterior located wound site on the limb, using the standard accessory limb model (ALM) surgery (*Figure 5A*; *Endo et al., 2004*; *McCusker and Gardiner, 2013*). Because the limb circumference of the small host animals was too small to receive the grafted blastema tissue from the large donors, we deviated the branchial nerve to a wound site on the flank of the host, where a larger skin wound could be made to fit the larger graft size (*Figure 5A*). The length of the ectopic limbs was measured bi- or tri-weekly. They were considered fully regenerated when the rate of growth of the ectopic limb was no longer significantly different from the uninjured limbs on the host animals. The donor animal limbs that were amputated to harvest the blastema grafts were also continually measured during regeneration as an additional control to measure the final sizes of the regenerated limbs if left on their native environment.

We hypothesized that if nerve abundance can regulate size of the limb regenerate, we would observe that the lengths of the grafted regenerates will correspond to the size of the host environment rather than the donor. Thus, blastemas from the small donors will produce large ectopic limbs when grafted to large hosts, and blastemas from large donors will produce small ectopic limbs when grafted to small hosts. Alternately, if nerve abundance does not influence regenerate size, then we would expect to see the blastemas from small donors on large hosts produce ectopic limbs smaller than the control grafts from large animals, and vice versa.

We observed that ALM environments with different nerve abundances altered both the growth rate and overall size of the regenerated limb structure. The blastemas (15 days post amputation) from small donors were initially 2-fold smaller than those from large donors when they were grafted onto the small and large hosts (*Figure 5C*, left panels, 0 day post graft [DPG]). Interestingly, we observed that the grafted tissues were the same size by 18 DPG within each host type and remained that way until regeneration was completed (*Figure 5—figure supplement 2A*). In comparison, the regenerated

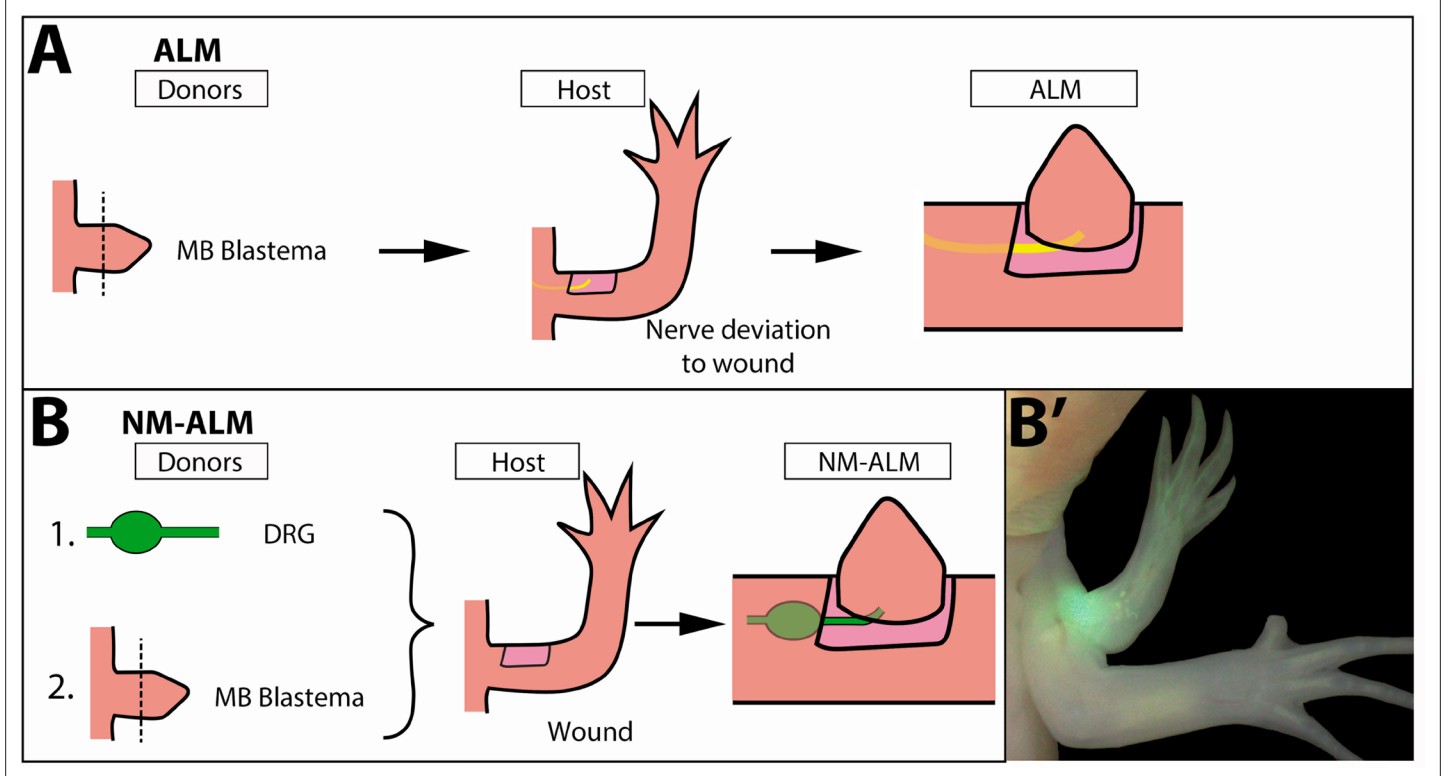

**Figure 6.** Decoupling host environment with nerve abundance using the neural-modified accessory limb model (NM-ALM). (**A**) The traditional ALM (as used in *Figure 5*) requires a blastema donor and a host animal with a nerve bundle deviated to the wound site. (**B**) The NM-ALM requires a GFP+ dorsal root ganglia (DRG) donor, blastema donor, and host limb with a wound site. (**B'**) The DRG's GFP+ axons regenerate and innervate the ectopic limb.

limbs that remained on the donor animals remained significantly different in size (*Figure 5D*). Although there was no significant difference in the size of regenerates that formed within an individual host type regardless of the origin of the blastema, we did observe significant differences when we compared regenerate sizes between the large and small hosts. By approximately 38 DPG, there was a significant difference in size between the ectopic limbs on the large and small host animals, and this continued until 130 DPG, when the growth rate of the grafted limbs matched that of the host limbs (*Figure 5C*, right panels; *Figure 5—figure supplement 2A*). This difference in size was due to a more rapid increase in the growth rate of the blastemas grafted to the large host animals (*Figure 5—figure supplement 2*). We hypothesize that the host environment on the large animals reaches a critical threshold of nerves to support growth at an earlier time and/or accumulates more nerve-dependent factors that promote growth in the regenerating tissues.

Together these data indicate that the growth rate and sizing of the limb regenerate positively correlates with nerve abundance in the host environment. However, it does not rule out other potential influences from the host environments that may also contribute to growth. Thus, we next designed an assay that decouples nerve abundance from the size of the host to test whether non-neural signals contribute to the sizing of the limb regenerate.

## Non-neural extrinsic signals in different sized hosts do not impact growth rate or size of the regenerated Limb

To evaluate the potential role of non-neural sources of growth regulation that may be present in the differently sized host animals, we developed a new regenerative assay called the NM-ALM that

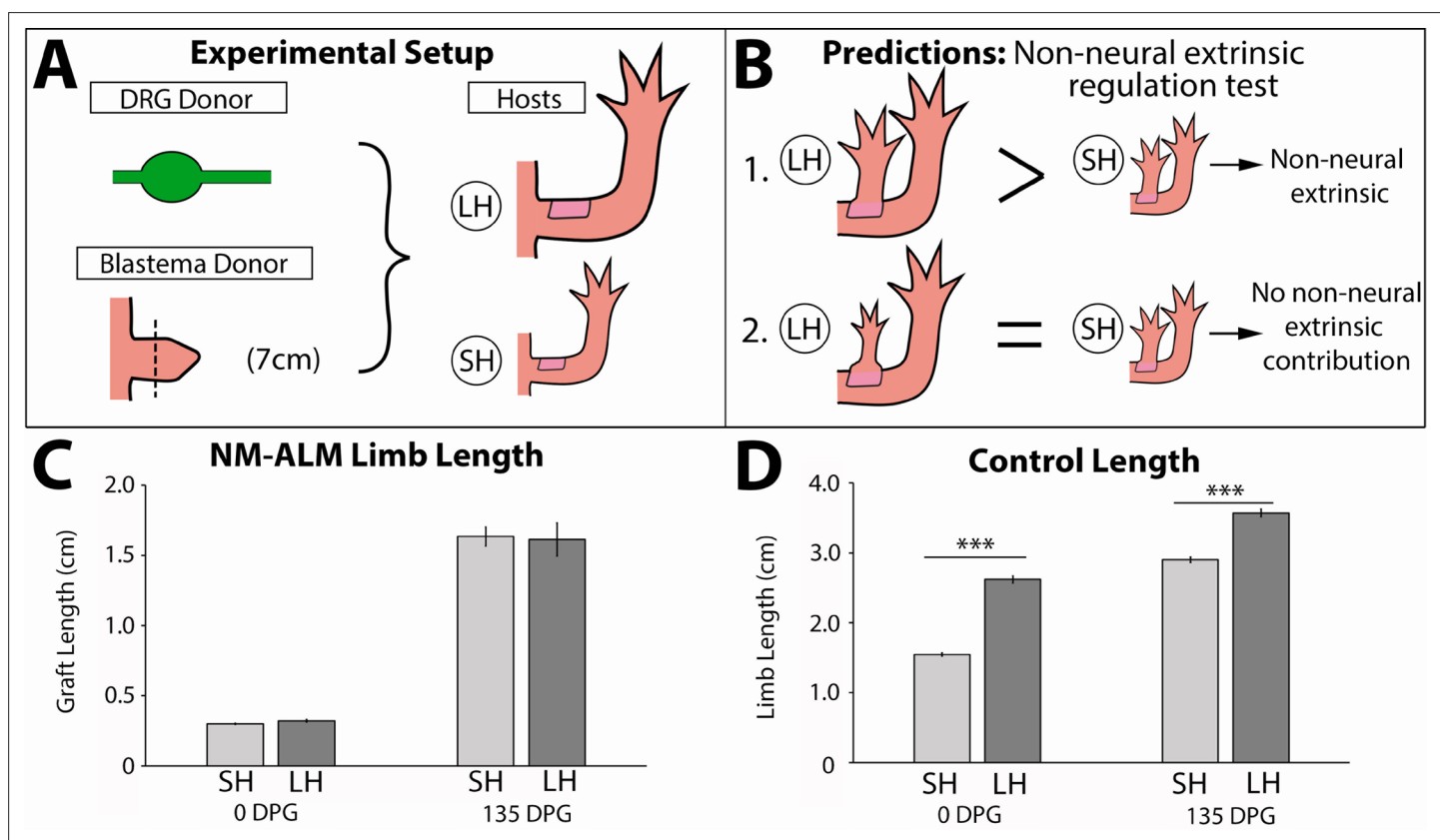

**Figure 7.** Regenerate scaling is not impacted by non-neural extrinsic factors in the neural-modified accessory limb model (NM-ALM). (**A**) Dorsal root ganglia (DRGs) from GFP+ donor animals (~14 cm) were grafted into wound sites on large (~14 cm, n = 12) and small (~7 cm, n = 19) host animals followed by mid-bud blastemas from (~7 cm) donor animals. (**B**) If non-neural extrinsic factors play an instructive role in size regulation, then the large host animals would produce a larger ectopic limb than those on the small host animal. If non-neural factors do not play an instructive role, then the ectopic limbs will be the same size, regardless of host size. (**C**) The ectopic limb lengths were the same size at 0 day post graft (DPG) and remained the same throughout regeneration (135 DPG). (**D**) The unamputated limbs on the control animals remained different sizes throughout the course of the experiment (n = 10). Error bars = SEM. p-Values calculated by t-tests. *** = p < 0.0005.

decouples the source of the nerves from the host environment (*Figure 6B*). In the NM-ALM, the forelimb-specific dorsal root ganglia (DRGs) from GFP+ donor animals are harvested and grafted into lateral limb wounds on host animals (one DRG per wound). The GFP+ DRGs are grafted below the mature skin next to the wound site, and the axon bundles are pulled into the middle of the wound site in a similar manner as the traditional ALM surgery. Mid-bud blastema staged regenerates from age-matched donors are then grafted onto the wound site (*Figure 6B*). We measured the length of the ectopic limbs in the NM-ALM weekly or bi-weekly, and their growth rates (cm/day) were compared to the unamputated limbs on the donors. The grafted limbs were considered to have completed limb regeneration when their growth rates were not significantly different from the growth rates of the unamputated limbs on the control animals. We observed that the implanted DRG supported the continued development of the blastema into a completely patterned and differentiated limb (*Figure 6B'*). Thus, the implanted DRGs are able to provide the appropriate signals to support the regenerative process.

To test whether non-neural extrinsic signals from the host environment provide cues that regulate the growth of the regenerating limb, we performed the NM-ALM on differently sized host animals (*Figure 7A*). If non-neural extrinsic signals play a role in regulating allometric growth of the regenerate, then we expected to observe that the regenerates that grew in the NM-ALMs on the large (14 cm) hosts would be larger in size than the ones that grew on the small (7 cm) hosts (*Figure 7B*). Alternately, if the limb nerves play the central regulatory role, we would not expect to see differences in the size of the regenerates on the different sized hosts. After 135 DPG, we observed that there continued to be no significant difference in growth rates and the length of the grafted regenerates

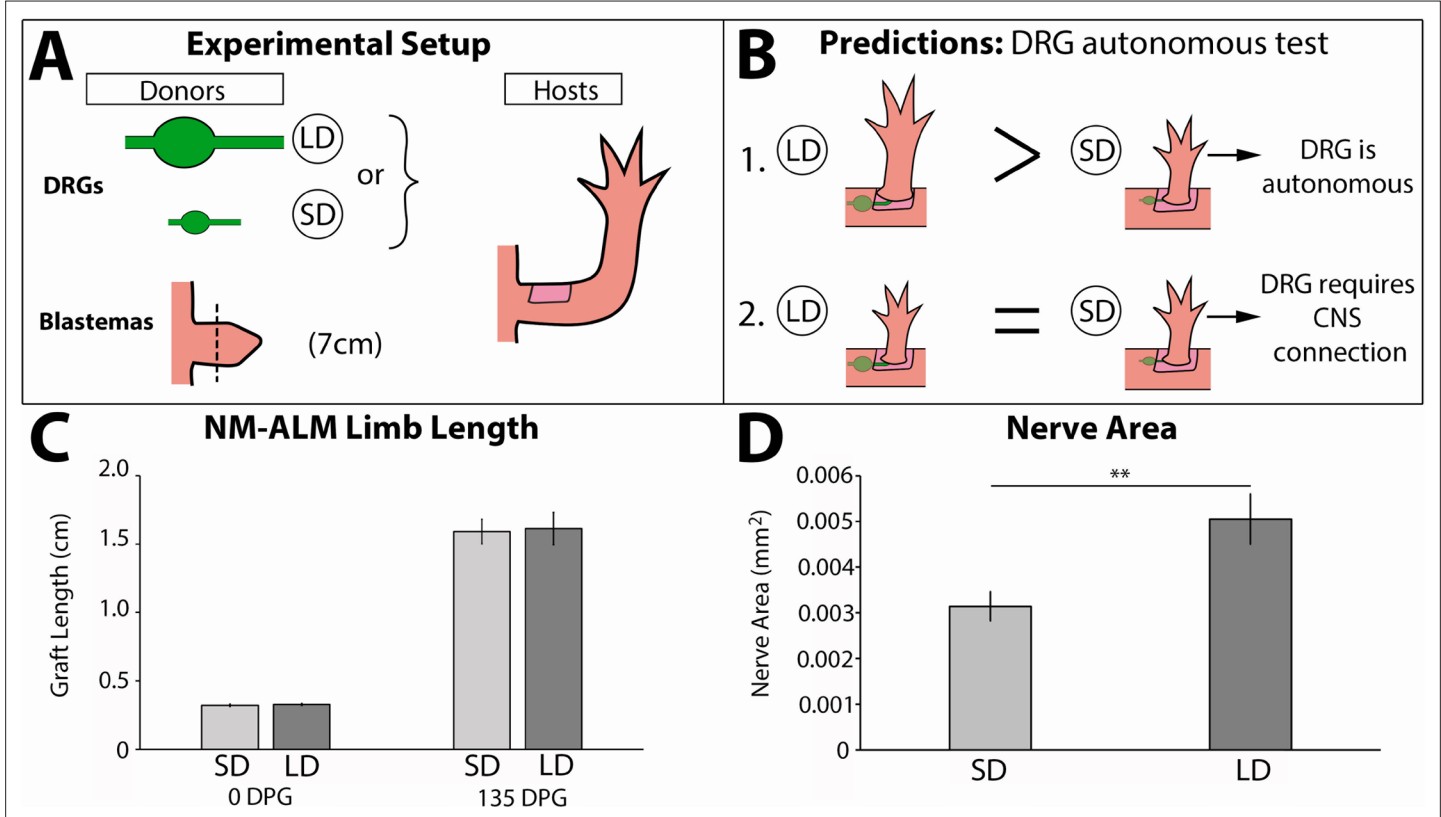

**Figure 8.** Neurons lose the ability to regulate allometric growth in the neural-modified accessory limb model (NM-ALM). (**A**) Dorsal root ganglia (DRGs) from large (~14 cm, n = 12) and small (~7 cm, n = 16) GFP+ animals were grafted into wound sites on host animals (~14 cm) followed by mid-bud blastemas from (~7 cm) donor animals. (**B**) If the ability to regulate size is autonomous to the DRGs, the DRGs from large animals will produce larger ectopic limbs than those from small animals. If size regulation is not autonomous, there will be no difference in ectopic limb size between grafts supplied by large or small animal DRGs. (**C**) The ectopic limb lengths were the same size at 0 day post graft (DPG) and remained the same throughout regeneration (135 DPG). Error bars = SEM. p-Values calculated by t-tests. (**D**) GFP+ innervation abundance was quantified in transverse sections of the ectopic limbs, showing significantly more nerve area (mm²) in the NM-ALMs grafted with DRGs from large donors compared to small donors (n = 5). Error bars = SEM. p-value = 0.0079, calculated by Mann-Whitney test.

on the different sized host animals (*Figure 5—figure supplement 2B* and *Figure 7C*). This trend was observed in NM-ALMs that were performed with DRGs that were harvested from both large and small animals (*Figure 5—figure supplement 2B*). In contrast, the control uninjured limbs on the small and large animals remained significantly different in size (*Figure 7D*). Because the size of the limb regenerate was not impacted by the host environment when the abundance of nerves was held constant, we conclude that non-neural extrinsic factors do not play a direct role in regulating allometric growth of the regenerating limb structure.

### Regulation of ontogenetic allometric growth is not autonomous to limb nerves

To test whether growth of the regenerate is autonomously regulated by the nerves, we leveraged our newly developed NM-ALM using DRGs that were harvested from different sized GFP+ donors (14 cm versus 7 cm) (*Figure 8A*). Our expectation was that if neural regulation occurs autonomously (at the DRG level), then the growth rate and overall size of the regenerated limb that grew on the DRG from the large donor would be larger than that which grew on the one from the small donor (*Figure 8B*). Conversely, if the DRGs require a connection with their native environment to regulate regenerate size, then we expected to see no difference in the regenerate growth rate and size regardless of the donor source of the DRG (large or small) (*Figure 8B*). Because we had previously observed a dose response in limb growth in partially denervated regenerating limbs (*Figure 4*), we hypothesized that allometric growth was regulated by nerve abundance alone. We quantified the abundance of innervation from the GFP+ DRGs in the ectopic limbs that grew from the NM-ALMs and found significantly more GFP+ neurons in the limbs with the large versus small DRG implants (*Figure 8D*). However, our data showed that there was no statistical difference in growth rates and the ectopic limb length between the regenerates from NM-ALMs with the DRGs from small or large animals (*Figure 5—figure supplement 2B* and *Figure 8C*). This trend was observed on the NM-ALMs on both the large and small hosts (*Figure 5—figure supplement 2B*). Thus, we concluded that nerve abundance alone does not regulate the final size of the regenerate and that factors from the DRGs' endogenous environment are required for the neural regulation of ontogenetic allometric growth during the tiny limb stage of regeneration. It is possible that without these upstream signals in NM-ALM, the DRGs from large and small donor animals generate the same amount of growth-promoting factors. Alternately, because the DRGs only contain sensory neurons, it is possible that motor neurons are required to control growth. Future studies will focus on the cells and signals that play a role in this upstream regulation.

## Discussion

One of the last essential steps of regeneration required to generate a fully functional limb is the growth of the regenerate to a size that is proportionally appropriate to the animal. However, little data had previously been collected on the post-blastema stages of regeneration and how allometric growth is regulated. This study constitutes the first thorough investigation of how the regenerating limb grows to the proportionally appropriate size. We have discovered that there are three distinct phases of growth prior to the limb completing regeneration: the blastema phase, the ETL phase, and the LTL phase (*Figure 1*). During the tiny limb phases of growth, the regenerate grows through increased cell proliferation, survival, and cell size (*Figure 2*), and limb nerves are required to maintain this growth (*Figures 4 and 5*). Our data indicates that limb nerves play a central role in the regulation of allometric growth (*Figures 5 and 7*), and that this regulation is not autonomous to the neurons but is somehow dependent on factors from the native neural environment (*Figure 8*).

### Regulation of growth and size in developing versus regenerating amphibian limbs

Previous studies on developing amphibian limbs suggest that both intrinsic and extrinsic factors determine size during embryogenesis. One example is from the classic cross-grafting study performed by Ross Harrison using limb buds between two differently sized salamander species (*Harrison, 1924*). In this experiment, limb buds from *Ambystoma tigrinum* (adult body length of 27 cm) were grafted to *Ambystoma punctatum* (adult body length of 16 cm, now known as *Ambystoma maculatum*) hosts and vice versa (*Harrison, 1924*). It was observed that the grafted limb buds grew to sizes that failed

to correspond with limbs from either donor or host. Rather, the limb buds from *A. punctatum* grew to a size that was smaller than both the host and donor limbs when grafted to *A. tigrinum,* while the *A. tigrinum* limb buds grew larger than the host and donor limbs when grafted to *A. punctatum*. Because the overall size of the grafted limb was influenced by both the host environment *and* the source of the graft, it was concluded that the growth of the limbs was regulated by both the intrinsic growth capacity of the limb bud cells and an extrinsic growth-promoting factor that is present in different abundances in the embryos of different host species (*Harrison, 1924*). It was later shown by Twitty and Schwind that when the host animals were fed to capacity, the growth rates of the grafted limb buds mimicked the donor animal's growth rate, but final size of the limb was more closely matched to the host (*Twitty and Schwind, 1931*). This observation suggests that the rate of allometric growth appears to be intrinsically regulated, while the duration of this growth is impacted more by extrinsic factors.

While our study also shows that extrinsic signals play a central role in the scaling of the regenerate, it is unknown whether the same nerve-dependent mechanisms are at play during limb bud development in the axolotl. During embryonic limb development, the apical ectodermal ridge (AER) signaling center is required for the production of paracrine factors such as FGFs and BMPs, which maintain an undifferentiated state and drive proliferation in the limb bud mesenchyme (*Purushothaman et al., 2019*; *Ovchinnikov et al., 2006*). The AER is established in an aneurogenic environment in multiple amphibian species including *Xenopus laevis*, *Rana pipiens*, and *A. punctatum/maculatum* (*Keenan and Beck, 2016*; *Kumar et al., 2011*; *Nieuwkoop and Faber, 1958*; *Taylor, 1943*; *Yntema, 1959*), while regenerating amphibian limbs require signaling from the nerve to establish the analogous structure, the apical epithelial cap (AEC) (*Singer, 1978*; *Singer and Inoue, 1964*). However, axolotl develop limb buds as larvae, not as embryos, and it is not clear whether the axolotl limb bud is innervated or not. Therefore, it is possible that the regulation of growth is dependent on nerve signaling in both embryonic and regenerating limbs.

It should be noted that the current study does not evaluate the influence of intrinsic factors that regulate the size of the regenerating limb. Because our study was performed on sibling *A. mexicanum*, there were minimal genetic differences between donor and host animals. Performing a similar cross grafting experiment, as was performed by *Harrison, 1924*, between species of different sizes in the context of regeneration would yield valuable insight in this regard.

## Mechanism of neurotrophic regulation of size during limb regeneration

Observations from multiple tetrapod species indicate that neural regulation of limb growth and size is a conserved mechanism. As we have explained previously, damage to the limb nerves in humans co-relates to decreased limb size, while overabundance of nerve signaling corresponds to limb or digit enlargement (*Bain et al., 2012*; *Cerrato et al., 2013*; *Labow et al., 2016*; *Tsuge and Ikuta, 1973*). The current study shows that the limb nerves also influence limb size in regenerating amphibians as well. This poses the amphibian limb regenerate as an exciting new model to study the general mechanisms underlying neural regulation of limb allometry.

Studies in developing embryos show that the alteration of factors that regulate patterning, elongation of the long bones, and cell physiology in the developing limb can all influence the final size of the limb. Interestingly, altering the expression of intrinsic factors, such as *Hox* transcription factors or paracrine factors such as *FGF*s, during the patterning or maturation stages of limb development, respectively, can both positively and negatively impact limb size (*Booker et al., 2016*; *Cho et al., 2008*; *Fromental-Ramain et al., 1996*; *Hérault et al., 1996*). In contrast, the alteration of factors that are required for the maintenance of cell physiology, such as DNA damage repair gene *Trp63*, only negatively impact limb size (*Vernersson Lindahl et al., 2013*). We interpret these observations to indicate that genes that regulate patterning or long bone growth are 'instructive' regarding the overall size of the limb, whereas genes that impact cell physiology play a more permissive role in this process. Whether and how the limb nerves regulate one or more of these different influencers of allometry during regeneration is an important outstanding question. However, it is interesting that the role of neural signaling appears to differ depending on the phase of growth. Limb denervation at the ETL phase negatively impacted proliferation, cell survival, and cell size, while the same manipulation at the LTL stage only impacted cell proliferation. It is unclear at this time whether the change in neural

requirement is due to alterations in the generation of growth-promoting signals from the nerve, the responsiveness of the regenerating tissue to these signals, or both.

While the molecular mechanism by which nerves regulate scaling has not been determined, the role of the limb nerves during the early stages of limb regeneration has been extensively studied and might be drawn upon to provide insight into their role in size regulation. Innervation of the wound epithelium during the early stages of limb regeneration is essential to establish the AEC. The AEC then produces growth factors essential to induce and maintain the cells in a dedifferentiated state and drive blastema cell proliferation (*McCusker et al., 2015*). *Satoh et al., 2016* demonstrated that FGF8 and BMP7 are directly produced by the nerves in the DRGs and migrate to the wound epithelium. Furthermore, regenerative failure caused by the denervation of the regenerating limb can be rescued by cocktails of growth factor proteins including FGFs and BMPs (*Makanae et al., 2014*; *Satoh et al., 2016*; *Vieira et al., 2019*), neuregulin-1 (*Farkas et al., 2016*), or AG (*Kumar et al., 2007*). Although the AEC is no longer present during the post-blastema stages of growth that we have focused on in the current study, nerves could both generate and induce the expression of similar growth-promoting factors in the late staged regenerating tissues. Alternately, or in conjunction with, the limb nerves may also regulate limb growth through their modulation of bioelectric signals which has been shown to regulate fin size in zebrafish (*Perathoner et al., 2014*; *Daane et al., 2018*). Regardless, the nature of the nerve-derived signals, and the downstream tissue and molecular targets of these signals, will be an essential focus of future research.

## Potential role of other extrinsic factors in determining size of the limb regenerate

The cessation of limb growth in determinately growing tetrapods, such as humans, is brought about by hormone-based changes which leads to alterations of chondrocyte behavior in the growth plate and completion of the terminal phase of differentiation in this tissue (reviewed in *Ağırdil, 2020*). In contrast, axolotl are an indeterminately growing species that maintain active growth plates in the long bones throughout life. Thus, the transition of positive to isometric ontogenetic allometric growth in the regenerated limb must occur without the closure of the axolotl growth plates and suggests an alternate mechanism of regulation. Our studies using the NM-ALM indicate that non-neural extrinsic factors, which would include systemic factors such as hormones, do not directly promote growth in the regenerating limb tissue, which further supports this idea (*Figure 7*). However, it is possible that systemic factors are permissive to regenerative growth. In determinately growing species, these factors play important roles in regulating the growth of long bones in preadult limbs (*Williams, 1981*; *Boersma and Wit, 1997*; *Penzo-Méndez and Stanger, 2015*).

Our observation that neurons, when separated from their typical environment within the vertebrae, are unable to support the positive allometric growth of the regenerate may indicate that additional signals that are extrinsic to the regenerating limb tissue are required for the neural regulation of scaling (*Figure 8* ad *Figure 5—figure supplement 2*). It is possible that either the vertebrae provide a permissive environment which allows the nerves to provide the appropriate cues to the regenerate, or there is some upstream signal that requires a direct connection of the limb nerves to the CNS to regulate growth. Alternately, because the NM-ALM assays that we performed only contain sensory neurons, it is possible that motor neurons can autonomously instruct growth in the regenerate. It will be important to resolve this issue in future studies.

## Conclusion

This study provides foundational knowledge on how proportion becomes reestablished in the regenerating limb in a continually growing system. Our data indicates that nerve signaling plays a central role in coordinating ontogenetic allometric growth, and future studies will focus on identifying the molecular mechanism(s) underlying this regulation. Furthermore, our data suggests that there is an upstream factor (or factors) required for the neural regulation of growth, potentially deriving from the endogenous environment of the limb nerves or the CNS. Lastly, as previously stated, our studies have not ruled out the likely intrinsic factors involved in size regulation. In total, there are still many unknowns, both up- and downstream of nerve signaling, that must be resolved to fully understand how allometric growth is regulated during axolotl limb regeneration. Furthermore, as regenerative medicine seeks to tap back into the developmental mechanism in order to regrow a fully functional

limb, it will be important to study size regulation in multiple species to identify the shared mechanisms regulating this process.

# Materials and methods

**Key resources table**

| Reagent type (species) or resource | Designation | Source or reference | Identifiers | Additional information |
|---|---|---|---|---|
| Genetic reagent (*Ambystoma mexicanum*) | White-strain 518 | Ambystoma Genetic Stock Center at the University of Kentucky | RRID: AGSC_101J | |
| Genetic reagent (*Ambystoma mexicanum*) | GFP-strain | Ambystoma Genetic Stock Center at the University of Kentucky | RRID: AGSC_110J | |
| Genetic reagent (*Ambystoma mexicanum*) | RFP-strain | Ambystoma Genetic Stock Center at the University of Kentucky | RRID: AGSC_112J | |
| Chemical compound, drug | MS-222 | Sigma-Aldrich | E10521-50G | |
| Chemical compound, drug | Formaldehyde | RICCA Chemical Company | R3190000-1A | |
| Other | DAPI | Sigma-Aldrich | D9542-5MG | (1:1000) dilution |
| Chemical compound, drug | Ethylenediaminetetraacetic acid (EDTA) | VWR | VWRV0105-500G | |
| Commercial assay or kit | In Situ Cell Death Detection Kit, Fluorescein | Roche | 11684795910 | |
| Commercial assay or kit | Click-iT Plus EdU Proliferation Kit | Roche | C10337 | 100 ng EdU/Animal |
| Software, algorithm | ZenPro software | Zeiss | 410138-1104-260 | |
| Software, algorithm | FIJI ImageJ | Open Source | | |
| Other | Tissue-Tek OCT Compound | Sakura | 4583 | |
| Other | Permount Mounting Medium | Thermo Fisher Scientific | SP15-100 | |
| Other | VECTASHIELD 612 Antifade Mounting Medium | Vector Laboratories | H-1000–10 | |
| Other | Wheat-Germ Agglutinin | Thermo Fisher Scientific | W32466 | |
| Other | Rhodamine phalloidin | Thermo Fisher Scientific | R415 | |
| Other | Hematoxylin Solution, Harris Modified | Sigma-Aldrich | HHS16-500ml | |
| Other | Eosin Y | Thermo Fisher Scientific | E511-100 | |
| Other | Alcian Blue | Sigma-Aldrich | A-5268 | |
| Antibody | (Mouse Monoclonal) Anti-Acetylated Tubulin antibody | Sigma-Aldrich | T7451-200UL | (1:200) dilution |
| Antibody | (Goat polyclonal) Anti-Mouse IgG Alexa Fluor 488 | Abcam | ab150173 | (1:200) dilution |

## Animal husbandry and surgeries

Ethical approval for this study was obtained from the Institutional Animal Care and Use Committee at the University of Massachusetts Boston (Protocol # IACUC2015004) and all experimental undertakings were conducted in accordance with the recommendations in the Guide for the Care and Use of Laboratory Animals of the National Institutes of Health. Axolotls (*A. mexicanum*) were spawned either at the University of Massachusetts Boston or at the Ambystoma Genetic Stock Center at the University of Kentucky. Experiments were performed on white-strain (RRID: AGSC_101J), GFP-strain (RRID: AGSC_110J), and RFP-strain (RRID: AGSC_112J) Mexican axolotls (*A. mexicanum*). Animal sizes are measured snout to tail tip and described in the text for each experiment. They were housed in 40% Holtfreters on a 14/10 hr light/dark cycle and fed ad libitum. Animals were fed every day or three times a week depending on the size of the animal. Animals were anesthetized in 0.1% MS222 prior to

surgery or imaging. Live images were obtained using a Zeiss Discovery V8 Stereomicroscope with an Axiocam 503 color camera and Zen software (Zeiss, Oberkochen, Germany).

To generate large and small animals, larval animals were either housed at 19°C or 4°C, which slows their growth rate. Animals were grown at these temperatures until their body lengths were approximately 2-fold different, at which point the smaller animals were moved to 19°C for 2 weeks prior to any surgical manipulation.

## Animal measurements

When measuring limb length and body length to determine limb proportionality and growth during regeneration, limbs were measured from the trunk/limb interception to the elbow and from the elbow to the longest digit tip (*Figure 1—figure supplement 1*). Body length was measured from snout to tail tip. All measurements were taken in centimeter (cm; *Figure 1—figure supplement 1*). Measurements were recorded prior to experimentation and weekly following surgical manipulation. After 5 weeks measurements were bi-weekly, and after 10 weeks they were taken tri-weekly.

## Limb amputations and staging of tiny limbs

Forelimb amputations are done mid-stylopod (mid-humerus). If the bone protruded from the amputation plane after contraction of the skin and muscle, it was trimmed back to make a flat amputation plane. Limb regeneration stages were determined through an observation of patterning and growth rates. Limbs were considered in the 'ETL' stage when they reached the mid-digit stage of patterning (*Iten and Bryant, 1973*). Prior to this, they are considered in the 'blastema' stage of limb regeneration. The transition from 'ETL' to 'LTL' is determined by a statistically significant decrease in limb length growth rate (cm/day). The regenerating limb is fully regenerated when the growth rate is no longer statistically significant from the unamputated control limbs.

## Limb denervation surgeries

Denervation of limbs was done by making a posterior incision on the flank, at the base of the arm, and severing and removing a piece of the three nerve bundles that come from spinal nerves 3, 4, and 5, proximal to the brachial plexus. A 2–3 mm piece of the nerve bundle was cut out in an effort to delay the regeneration of the nerves into the regenerates. Partial denervations were also performed by severing 1, 2, or all three of the limb nerve bundles.

denervations were performed by severing spinal nerve 5. 2/3 denervations were performed by severing spinal nerves 4 and 5. Full denervations were performed by severing all three nerves. Mock denervations were also performed by creating the same incision on the posterior side of the limb and dissecting the nerve bundles as in a typical denervation but leaving the nerve bundles intact. Because limb denervation last approximately 7 days before nerves begin to re-innervate the limb, experimental limbs were analyzed 4 days post denervation.

## ALM/blastema grafting

Limbs on large (average 12 cm snout to tail tip) and small (average 6 cm snout to tail tip) GFP+ donor axolotls were amputated mid-stylopod and allowed to regenerate until mid-bud stage. At that stage, they were amputated along with approximately 2 mm stump tissue, and grafted onto a regenerative permissive environment on large (12 cm snout to tail tip) RFP+ host axolotls or small (6 cm snout to tail tip) white host axolotls. Stump tissue was included in the graft to ensure the ectopic limb is composed primarily of donor animal cells, since stump cells contribute to the regenerate (*Currie et al., 2016*). The regenerative permissive environment on the large hosts was created by removing an anterior patch of full thickness skin from the stylopod and deviating a limb nerve bundle to the wound site (*Endo et al., 2004*; *McCusker and Gardiner, 2013*). The blastemas on the large donor animals were substantially larger than the limbs of the small host animals, making it impossible to perform this test on the small limbs. Thus, a regenerative permissive environment on the small host axolotl was generated by removing a patch of full thickness skin from flank of the animal posterior to the forelimb. The limb nerve bundle is dissected from the limb and deviated to the flank wound site on the small animal. After the blastemas are grafted on to the host wound sites, hosts are kept on ice and misted frequently for 1 hr, permitting attachment of the graft.

## Neural-modified ALM

We developed the NM-ALM assay to determine if non-neural extrinsic factors were contributing to size regulation. In the NM-ALM the nerve source and host environment are decoupled by grafting a limb DRG from a donor animal in lieu of the deviated nerve bundle from the host animal into the wound site, as in the standard ALM surgery. Anterior wounds are created on white-strain host animals. Limb DRGs were carefully extracted postmortem from GFP+ donor animals and implanted below the proximal skin surrounding the wound site (*Figure 6A*). The limb axon bundle is positioned in the middle of the wound site (*Figure 6A*). Mid-bud staged blastemas, along with approximately 2 mm of stump tissue, were amputated from white-strain donor animals and immediately grafted onto the DRG nerve bundle and wound site on the host animals. Animals were kept on ice and moist for 2 hr following grafting to ensure attachment of the graft.

In the NM-ALM experiment reported here, we used large (average 14 cm snout to tail tip) and small (average 7 cm snout to tail tip) GFP+ and white-strain siblings, generated by crossing heterozygous GFP parents, as blastema donors, DRG donors, and NM-ALM hosts. DRGs from both large and small GFP+ donor animals were dissected postmortem and grafted into anterior limb wound sites on large and small white host animals (*Figure 6C*). Mid-bud staged blastemas from small white-strain donors were then grafted onto the NM-ALM.

## Tissue histology and immunofluorescence

Tissues were fixed overnight at 4°C in 4% formaldehyde (RICCA Chemical Company, Arlington, TX), decalcified in 10% EDTA (VWR, Radnor, PA) for 5–14 days depending on tissue size, rehydrated in 30% sucrose for 2 days, and embedded in Tissue-Tek OCT Compound (Sakura, Torrance, CA). The OCT blocks were then flash-frozen in liquid nitrogen and stored at –20°C. They were cryo-sectioned on a Leica CM 1950 (Leica Biosystems, Buffalo Grove, IL) through the mid-zeugopod at 7 µm thickness. Following staining, histological stained slides were mounted using Permount Mounting Medium (Thermo Fisher Scientific, Waltham, MA). VECTASHIELD Antifade Mounting Medium (Vector Laboratories, Burlingame, CA) was used to mount coverslips on the fluorescently stained slides. Sections were imaged on the Zeiss Observer.Z1 at 20× magnification. Tile scans were taken of the entire tissue section and then stitched together using the ZenPro software (Zeiss).

## Quantification of cell proliferation and cell death

For analysis of cell proliferation in the limbs, 100 ng of EdU (Roche, Basel, Switzerland) was injected into the intraperitoneal space on the flank of the animal. The limbs were then collected exactly 4 hr after injection and prepared for staining. Harvested tissues were process for cryo-sectioning as described above. The Roche Click-It EdU kit was used, following manufacturer's protocol, and co-stained with DAPI (1:1000 dilution – Sigma-Aldrich, St Louis, MO). The Roche In Situ cell death detection kit (Fluorescein) was used to analyze apoptosis cell death using the manufacturer's protocol. TUNEL-stained sections were also co-stained with DAPI to obtain percent cell death. Using the Open Source FIJI software, the number of either proliferating (EdU+) or dying (TUNEL+) cells and DAPI+ cells was counted and used to calculate the labeling indices for each. In each section, the tissues were visually separated based on morphology into three basic categories: epidermal, skeletal (bone or cartilage), and soft tissue (all other tissue), and the labeling indices were generated for each separately. Three technical (three sections per limb) and at least three biological replicates were performed for each sample.

## Quantification of cell size and ECM size

The quantification of average cell size and extracellular area were performed separately on the entire epidermal, muscle, and skeletal tissues in each tissue section. The epidermal and muscle tissues were identified based on morphology in WGA (Thermo Fisher Scientific), Rhodamine phalloidin (Thermo Fisher Scientific), and DAPI (1:1000 dilution – Sigma-Aldrich) stained sections. Skeletal tissues (bone or cartilage) were identified in sections stained with Harris Hematoxylin (Sigma-Aldrich), Eosin Y (Thermo Fisher Scientific), and Alcian Blue (Sigma-Aldrich). To measure the average cell size of epidermis and muscle in the fluorescent images, the area within WGA plasma membrane-stained cells, where the nucleus was observed (DAPI signal), was quantified using the FIJI software and averaged (*Figure 2— figure supplement 1*). While muscle cells are elongated syncytial cells, our measurements only quantified a cross-sectional area. For the skeletal elements, the average cell area was quantified by

measuring the area of Alcian blue negative areas in the element that contained a nucleus (*Figure 2—figure supplement 1*).

To analyze ECM size for the epidermis and muscle, the area of all WGA internal cell spaces (with and without DAPI) were quantified. For skeletal tissue, the area of the Alcian blue negative cell spaces were quantified. The sum of these areas was calculated to obtain the 'total cellular area'. The area of the complete tissue 'total tissue area' was then quantified (*Figure 2—figure supplement 1B*). Since the tissue sizes can vary, percent ECM was determined through the following equation:

$$\text{Percent ECM Deposition} = \frac{\text{Total Tissue Area} - \text{Total Cellular Area}}{\text{Total Tissue Area}} \times 100$$

## Neural staining

The quantitative analysis of nerve abundance was performed on regenerating limb and flank sections using the Mouse Monoclonal Anti-Acetylated Tubulin antibody (1:200 dilution – Sigma-Aldrich), followed by the Goat-Anti-Mouse IgG Alexa Fluor 488 (1:200 dilution – Abcam, Cambridge, MA) secondary antibody. They were co-stained with Rhodamine phalloidin and DAPI as a general tissue stain, and to provide positional context for the location of the axon bundles. Limb nerve abundance was quantified by determining the percentage of limb area that is positive for Anti-Acetylated Tubulin staining. To determine limb-bound axon bundle size, the sum of the area of axon bundles from DRGs 3, 4, and 5 were quantified as they emerged from the skeletal muscle surrounding the spine (*Figure 5—figure supplement 1*).

## Acknowledgements

The authors wish to thank Dr Kellee Siegfried Harris, her lab members, and the members of the McCusker lab for their insightful comments during the development of this project.

## Additional information

### Funding

| Funder | Grant reference number | Author |
|---|---|---|
| National Institutes of Health | 0R15HD092180-01A1 | Catherine D McCusker |
| University of Massachusetts Boston | Doctoral Dissertation Grant | Kaylee M Wells |

The funders had no role in study design, data collection and interpretation, or the decision to submit the work for publication.

### Author contributions

Kaylee M Wells, Conceptualization, Data curation, Formal analysis, Funding acquisition, Investigation, Methodology, Project administration, Validation, Visualization, Writing - original draft, Writing - review and editing; Kristina Kelley, Data curation, Formal analysis, Investigation, Validation, Visualization, Writing - review and editing; Mary Baumel, Data curation, Formal analysis, Methodology, Validation, Visualization, Writing - review and editing; Warren A Vieira, Conceptualization, Formal analysis, Investigation, Methodology, Writing - review and editing; Catherine D McCusker, Conceptualization, Funding acquisition, Methodology, Project administration, Resources, Supervision, Visualization, Writing - original draft, Writing - review and editing

### Author ORCIDs

Mary Baumel http://orcid.org/0000-0002-8551-4785
Catherine D McCusker http://orcid.org/0000-0003-0127-433X

### Ethics

This study was carried out in accordance with the recommendations in the Guide for the Care and Use of Laboratory Animals of the National Institutes of Health. The experimental work was approved by

the Institutional Animal Care and Use Committee of the University of Massachusetts Boston; protocol number IACUC2015004, animal welfare assurance number D16-00246 (A3383-01).

### Decision letter and Author response

Decision letter https://doi.org/10.7554/eLife.68584.sa1
Author response https://doi.org/10.7554/eLife.68584.sa2

## Additional files

### Supplementary files
• Transparent reporting form

### Data availability
The raw data used to generate the figures for this paper are available in the source data files corresponding to that figure.

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
