## [Editor Report]

It has long been known that nerves regulate the early formation of the blastema during limb regeneration through the promotion of cell proliferation. The manuscript provides an interesting new role for nerves during salamander limb regeneration by showing that nerves also determine how much tissue to regenerate. They demonstrate that increased nerve abundance makes bigger limbs while a decrease in nerve abundance generates smaller limbs. Size regulation of organs is a broadly interesting and clinically important problem, which is why this manuscript should be of interest to a large general audience.

---

## [Decision Letter]

**Decision letter after peer review:**

Thank you for submitting your article "Neurotrophic control of size regulation during axolotl limb regeneration" for consideration by *eLife*. Your article has been reviewed by 2 peer reviewers, and the evaluation has been overseen by K VijayRaghavan as the Senior and Reviewing Editor. The following individual involved in the review of your submission has agreed to reveal their identity: Matthew Harris (Reviewer #2).

Essential revisions:

As the authors will see from the reviews below, there are important changes that address conveying the results better, and also in the Introduction and in the Discussion. These can be readily and speedily addressed.

*Reviewer #1 (Recommendations for the authors):*

An overview of the limb regeneration stages would be helpful in the introduction. I recall that there are several classifications that exist in the literature with some stating 2 phases, 3 phases, or even 4 phases? I wrote this before getting to the results so I know it is discussed, but as a reader, I thought it missing in the introduction.

Line 57-59 – Could you clarify why Harrison and Twitty concluded intrinsic and extrinsic mechanisms? I thought the conclusion based upon balancer graft between species and tiger/maculatum grafts (?) concluded an intrinsic mechanism.

The introduction seems to be missing an explanation for how nerves communicate with the limb. For example, it would be interesting to the reader to know that it is not the type of nerve that impacts (early) growth and that it is a protein rather than a neurotransmitter. It is not known, though that the mechanism described here uses the same mechanisms, but I feel it would provide more context in the introduction.

Line 69-71 – I didn't know that neurofibromas increase digit size…interesting.

Figure 1B – I don't see the error bars in figure 1B.

One of the most interesting points for future studies is to determine if motor or sensory nerves have differential impacts on size regulation of the limb regenerate. Is the lack of growth because motor nerves are not innervating muscles, which then degenerate and lead to a lack of growth? Or is it that the impact is general and growth is inhibited by the loss of nerve axon secretions? Yet, the experiments in Figure 6 suggest that sensory nerves are all that are needed for regeneration.

Line 166 – The sentence references Supplemental Figure 2, but I think it should be Supp Figure 3.

Figure 4. One consideration that may be worth mentioning is that the amount of necrosis is unknown after denervation. The authors looked at apoptosis using TUNEL, but it is possible that a significant amount of necrosis also occurs.

Line 266: The housing of animals at different temperatures is interesting and impressive that a 2 fold difference in size was accomplished.

Figure 6: Do nerves from the host animal invade the accessory limb in the NM-ALM model? This is important for considering if the interactions of motor nerves play any role in the regulation of growth control. I am not suggesting to run the experiment, but it may be worth discussing.

Line 351: What are the extrinsic signals that may be regulating growth size. I suggest taking the term (and likely) out here and just bring it up in the discussion.

Figure 8: There is an alternative conclusion that should be considered here. Even though the DRG is smaller, is it possible that it makes an equivalent amount of mitogenic factors as larger DRGs? Or is it possible that the smaller DRG has more growth potential so can make more abundant close contacts with cells than the larger DRGs? I think a sentence or two explaining alternative conclusions can be included.

Line 393: There is an extra "the" in the line.

Line 406: There is now strong evidence in the axolotl developing limb that the epithelium does not express FGFs. I think a caveat should be included mentioning the work from Purushothaman et al., 2019 that shows FGF expression is mesenchymal during axolotl limb development.

*Reviewer #2 (Recommendations for the authors):*

Experimental tissue-level approaches as defined in this work, while lacking in molecular detail, often have a broader lasting impact in the field than the latter: AER and ZPA extirpation and transplant studies are still being investigated to understand the organizational potential of these tissues even after the discovery of FGF and SHH mediated signaling. This study has the potential to be a landmark paper providing the foundations in which other molecular studies will follow. However, the paper needs some critical refinement in the precision of definition and defining the biological mechanisms of the process under study (catch up growth? Size-determination?). Even for a reader who appreciates these questions and is familiar with the current understanding of regulation of proportion, I had a hard time distilling the key message of the paper and the implications of the findings. The limitation lies not within the experiments or in their immediate analysis, rather integration of the model and experimental findings into a cohesive whole. I think my major criticisms can be accommodated by textual changes and distilling the overall message. I have broken up comments into general areas that I feel would benefit attention in review.

Description of the problem and biological process:

One of the major difficulties I had was understanding the importance of the tiny limb and what biological process the authors were in fact studying. In part this is due to an introduction that was not focused on the broad topic of how rate of growth, or its cessation, is modulated to accommodate for proportionality of size. It is unclear if the authors are also arguing the determination of size itself as a set point of size, or if the mechanisms is simply modulation or rate, or both. Secondly, it should be made clear very early why their delineation of multiple phases of growth is an important and novel perspective; as such how it now can help elucidate the mechanisms underlying this regulation.

Nerve activity and relative size:

There are several areas that need to be addressed concerning description and discussion of nerve activity and size. One is strictly terminology of "nerve abundance" or "innervation" when in fact it is not really known what the innervation extent is within the limb – just the size of the host nerve. Thus the correlative finding is to relative contribution, or extent, of nerves provided to the regenerate – or within the regenerate as detailed in Figure 3. In Figure 3. it will be important to show absolute levels of nerve abundance (or normalized to body length), as well as the change, looks predominantly to be due to the addition of non-neural tissue leading change in of cross-sectional volume of the regenerate -- reduction of total mass of the early tiny limb compared to more mature limb will increase the fraction of nerve density but the absolute innervation is constant. Is then growth regulation titering out of nerve-derived signals? This seems to be the argument, but is phrased as relative levels of nerve innervation or abundance – or even hyper-innervated, which seems to imply more nerves are entering or anastomosing in the limb in these stages. This has not been shown. If that is the specific statement the authors want to use, it begs the question if one raises innervation (added DRG graphs or extra nerve re-routing to a regenerative stump e.g. branchial plexus being diverted to the limb), can this extend growth of a regenerate through modulation of growth rate at the tiny limb phase??

One more critical area is the description of nerve instructional information arising from nerve 'abundance' from Figure 5. This figure was hard for me to understand – in part due to titles of Panel 5c and 5d it was not clear what the difference between Graft Tissue Size and Donor Size is. But, beyond that the inference is that the larger animals have larger nerves at the stump (innervation abundance of the title) thereby being causative of the change. However, the findings are that host size determine size of the graft. The dependency of the nerve in this instruction is inferred by the requirement of the nerve for regenerative growth – but as the NM_ALM model with different sized axonal bundles shows (Figure 8), it is not instructional (at least when neural explants are used). It would be good if the authors could refine their language on abundance and innervation to be more precise on what is being measured in the experiment.

If the argument is really the amount of innervation, couldn't several DRGs be placed in a single wound site to assess this directly? Can a larger limb regenerate over the contralateral limb? There might be other reason this experiment is not feasible, but raises some interesting support for the authors argument. Note L84 "Our data shows // that changes in the nerve abundance are sufficient to manipulate (positively or negatively) the ultimate size of the limb regenerate." I don't know if the authors have shown positive regulation formally.

Note: p475 The authors mention humans with 'decreased limb size, while an overabundance of innervation corresponds to limb or digit enlargement' is the case in some cases of macrodactyly However, in cases of macrodactyly or overgrowth can also be caused by harmatomas (e.g. NF1) leading to altered nerve activity not innervation amount per se. Note, this is consistent with the growth factor production argument of the authors that follows in the text.

Patterns of Growth rate:

Near the end of the paper, the authors present growth rate data over time for the regenerate of short limb and long limb groups. Figure S7 is really quite informative in demonstrating a shift in the rate of growth seen in Small hosts over that in Large hosts; the general rate of growth of regenerates from smaller hosts is initiated later -- this seems to be the crux of the difference. The result is that the limb never catches up to the size of limbs with small hosts but grows at generally the same rate. Could the authors integrate their findings with this latent initiation and nerve activity of the stump? Levels of nerve growth factors have been shown to be sufficient to replace nerve dependence – could it be that during early regeneration large hosts have higher neurotropic growth factor levels and this triggers a threshold-dependent growth program; small hosts would require longer time to reach this threshold and initiate development.

Suggestions

Figure 4. Could examples of histology reflecting the differences in the bar plotz of Cell proliferation/Cell death/ and size be shown as small insets. This will also clarify what it is that is being measured and what staining might look like. I saw some of the metrics in the Supplementary files but seeing the histology or cell death signals would help in understanding your data in the figure proper.

General editorial comments.

Might consider deleting NS in graphs where the statistical significance failed to pass p<0.05 threshold. It will make the graphs easier to read.

Overall, the paper is quite a unique contribution to the field and would be fitting for the *eLife* audience. However, the impact of this manuscript is dependent on refining the argument such that it is clear what specifically is being shown and it direct significance in trying to understand the regulation of proportion.

---

## [Author Response]

Essential revisions:As the authors will see from the reviews below, there are important changes that address conveying the results better, and also in the Introduction and in the Discussion. These can be readily and speedily addressed.Reviewer #1 (Recommendations for the authors):An overview of the limb regeneration stages would be helpful in the introduction. I recall that there are several classifications that exist in the literature with some stating 2 phases, 3 phases, or even 4 phases? I wrote this before getting to the results so I know it is discussed, but as a reader, I thought it missing in the introduction.

Reviewer 1 requested that we include additional background in the introduction including (1) An overview of the limb regeneration stages would be helpful in the introduction, (2) clarification of our description of the Harrison and Twitty studies, and (3) further description of what is known about of neural communication in the limb. We have added and altered text both in the introduction as well as the Discussion sections to address this issue.

Line 57-59 – Could you clarify why Harrison and Twitty concluded intrinsic and extrinsic mechanisms? I thought the conclusion based upon balancer graft between species and tiger/maculatum grafts (?) concluded an intrinsic mechanism.The introduction seems to be missing an explanation for how nerves communicate with the limb. For example, it would be interesting to the reader to know that it is not the type of nerve that impacts (early) growth and that it is a protein rather than a neurotransmitter. It is not known, though that the mechanism described here uses the same mechanisms, but I feel it would provide more context in the introduction.Line 69-71 – I didn't know that neurofibromas increase digit size…interesting.Figure 1B – I don't see the error bars in figure 1B.One of the most interesting points for future studies is to determine if motor or sensory nerves have differential impacts on size regulation of the limb regenerate. Is the lack of growth because motor nerves are not innervating muscles, which then degenerate and lead to a lack of growth? Or is it that the impact is general and growth is inhibited by the loss of nerve axon secretions? Yet, the experiments in Figure 6 suggest that sensory nerves are all that are needed for regeneration.

Reviewer 1 thought that “one of the most interesting points for future studies is to determine if motor or sensory nerves have differential impacts on size regulation of the limb regenerate. Is the lack of growth because motor nerves are not innervating muscles, which then degenerate and lead to a lack of growth? Or is it that the impact is general and growth is inhibited by the loss of nerve axon secretions?” This is a very interesting point. Historic studies on blastema development have shown that's signals from either motor or sensory neurons are sufficient to support regeneration. The NM-ALM studies we have reported here also support this, since a complete regenerate forms on an implanted DRG. However, our current study has not thoroughly evaluated the different potential inputs form both sensory and motor neurons in the regulation of limb size. We are also very interested in this issue, and this will be an important issue to resolve in future studies. As the reviewer also pointed out, there are potential tissue-specific interactions between different neuron types, thus it will be important to include tissue specific analyses in the future as well. However, we should note that although degeneration of muscle would likely result in decreased girth of the limb, it is less likely to affect the overall length of the limb, which is what we have quantified in the current study. We think that there are two possible mechanisms; either the neurons secrete factors that directly regulate the overall growth of all tissues of the late staged regenerate; or the neurons regulate the growth of a specific tissue (potentially the skeletal tissue), which then regulates the growth of the other tissues. This is an important question which we are actively trying to answer. We have discussed these open questions in more detail in the Discussion section.

Line 166 – The sentence references Supplemental Figure 2, but I think it should be Supp Figure 3.Figure 4. One consideration that may be worth mentioning is that the amount of necrosis is unknown after denervation. The authors looked at apoptosis using TUNEL, but it is possible that a significant amount of necrosis also occurs.

In reference to Figure 4, Reviewer 1 requested that we consider that necrosis may also occur in the limb tissue after denervation. Because TUNEL staining detects double stranded breaks, which occur in both necrotic and apoptotic cells, we are unable to decipher from this assay between these two possible mechanisms of cell death. We have made sure to now describe the phenomenon that we are observing as “cell death” so to not inappropriately assign a specific cell death mechanism.

Line 266: The housing of animals at different temperatures is interesting and impressive that a 2 fold difference in size was accomplished.Figure 6: Do nerves from the host animal invade the accessory limb in the NM-ALM model? This is important for considering if the interactions of motor nerves play any role in the regulation of growth control. I am not suggesting to run the experiment, but it may be worth discussing.

In reference to Figure 6, Reviewer 1 asked whether nerves from the host animal invade the accessory limb in the NM-ALM model. “This is important for considering if the interactions of motor nerves play any role in the regulation of growth control.” In the resubmitted manuscript we have provided a new panel in Figure 8 that quantifies nerve abundance when DRGs of different sizes are grafted into the NM-ALM. As you have pointed out, it is possible that motor neurons from the host animals may have invaded the accessory limb. When analyzing innervation in the NM-ALM limbs we also used a general neural stain, and did not find the contribution of non-GFP+ (host origin) axons in the ectopic limbs.

Line 351: What are the extrinsic signals that may be regulating growth size. I suggest taking the term (and likely) out here and just bring it up in the discussion.Figure 8: There is an alternative conclusion that should be considered here. Even though the DRG is smaller, is it possible that it makes an equivalent amount of mitogenic factors as larger DRGs? Or is it possible that the smaller DRG has more growth potential so can make more abundant close contacts with cells than the larger DRGs? I think a sentence or two explaining alternative conclusions can be included.

In reference to Figure 8, reviewer 1 wanted us to consider alternate conclusions. “Even though the DRG is smaller, is it possible that it makes an equivalent amount of mitogenic factors as larger DRGs? Or is it possible that the smaller DRG has more growth potential so can make more abundant close contacts with cells than the larger DRGs?” These are interesting interpretations which we have now included in the text. By analyzing the abundance of innervation in the NM-ALM induced limbs, we found that there is significantly greater innervation in the limbs with the larger DRG, indicating that even if the small DRG has more neuronal growth potential, that it is not enough to surpass the large DGR in terms of innervation of the regenerating tissue. However, as Reviewer 1 pointed out, it is possible that the small DRG is somehow generating the same abundance of mitogenic factors as the large DRG. We note that in the experiment in Figure 5, the grafts onto the small hosts (with the small nerve source) grow a smaller size than those grafted to large host animals. Thus, in the ALM assay, any possible impact of nerve size on the generation of mitogenic factors or neural growth in the small animals are not enough to reach the size of the grafts on the larger hosts. However, it is possible that in the NM-ALM, where the nerves are removed from their native environment, that the small DRGs may adapt to the new environment better than the large DRG grafts. When we identify the factor(s) that the DRG generated to promote growth in future experiments, this will be one of the first questions that we want to answer.

Reviewer #2 (Recommendations for the authors):Experimental tissue-level approaches as defined in this work, while lacking in molecular detail, often have a broader lasting impact in the field than the latter: AER and ZPA extirpation and transplant studies are still being investigated to understand the organizational potential of these tissues even after the discovery of FGF and SHH mediated signaling. This study has the potential to be a landmark paper providing the foundations in which other molecular studies will follow. However, the paper needs some critical refinement in the precision of definition and defining the biological mechanisms of the process under study (catch up growth? Size-determination?). Even for a reader who appreciates these questions and is familiar with the current understanding of regulation of proportion, I had a hard time distilling the key message of the paper and the implications of the findings. The limitation lies not within the experiments or in their immediate analysis, rather integration of the model and experimental findings into a cohesive whole. I think my major criticisms can be accommodated by textual changes and distilling the overall message. I have broken up comments into general areas that I feel would benefit attention in review.

A major criticism from Reviewer 2 was that “the paper needs some critical refinement in the precision of definition and defining the biological mechanisms of the process under study (catch up growth? Size-determination?).” Because we are measuring the growth of the regenerating structure relative to growth of the rest of the body in individual axolotl, we feel that the best description of this is ontogenetic allometric growth. The regulation of this growth will determine the overall size of the regenerated structure once its growth is isometric with the rest of the animal. We do not think that the growth of the tiny limb is a form of catch-up growth because the growth if the regenerate is hyperallometric and independent of the stage of life, whereas catch-up growth is isometric and requires the animal to be in a pre-adult stage of life. We had originally discussed both catch up growth and compensatory regeneration in the Discussion section, however we have removed it because we felt it may be a distraction to the readers.

Studies in mutant mammalian embryos have shown that proportionality of limb size can be positively and negatively impacted by altering factors that play a role in either limb patterning (ex *Hoxs*) or the elongation of the skeletal bones (ex *Ihh*). Thus, size determination of the developing tetrapod limb appears to be regulated at these two different stages of limb formation. Since the regenerating tiny limb has completed patterning, the regulation of size at this stage would appear to be synonymous with the latter stage of limb development in embryos, where the maturing limb structures elongate. We have added these points to the discussion.

Description of the problem and biological process:One of the major difficulties I had was understanding the importance of the tiny limb and what biological process the authors were in fact studying. In part this is due to an introduction that was not focused on the broad topic of how rate of growth, or its cessation, is modulated to accommodate for proportionality of size. It is unclear if the authors are also arguing the determination of size itself as a set point of size, or if the mechanisms is simply modulation or rate, or both. Secondly, it should be made clear very early why their delineation of multiple phases of growth is an important and novel perspective; as such how it now can help elucidate the mechanisms underlying this regulation.

One of the major difficulties that Reviewer 2 had “was understanding the importance of the tiny limb and what biological process we were studying. In part this is due to an introduction that was not focused on the broad topic of how rate of growth, or its cessation, is modulated to accommodate for proportionality of size. It is unclear if the authors are also arguing the determination of size itself as a set point of size, or if the mechanisms is simply modulation or rate, or both. Secondly, it should be made clear very early why their delineation of multiple phases of growth is an important and novel perspective; as such how it now can help elucidate the mechanisms underlying this regulation.”

The suggestion to include a more through explanation of how rate of growth is modulated to control proportionality of size is well-heard, and we have included this as new text in the introduction. Our data indicates that proportionally of size in the tiny limb is regulated by both an increase in growth rate that is disproportionate to the rest of the animal, and the duration of this growth. As we have explained above, we think that the best description of this process is ontogenetic allometric growth. Because the axolotl grows indefinitely, the growth of the regenerated limb does not completely cease, but rather, slows to the growth rate of an uninjured limb (isometric growth). Thus, in the case of axolotl limb regenerate, this argues against a “set point” of limb size.

To our knowledge, all of the examples of allometric growth in tetrapod limbs seem to occur as a single phase, although whether this growth exhibits positive or negative allometry depends on the limb and the species. For example, bat forelimbs exhibit a strongly positive allometric growth relative to the rest of the developing bat body, which does not occur in mice (Booker et al., 2016). Thus, our observation that growth of the regenerating axolotl limb is biphasic appears to be unique to post-embryonic limb development. Although the underlying cause and reason for this difference in growth rate are unknown, we have discovered that the role of neurons in regulating these phases is different. Innervation at the ETL stage is required for increased cell proliferation, decreased cell death, and increased cell size, while at the LTL stage, innervation appears to be just required for cell proliferation. We speculate that this difference in dependency is due to a change in tissue maturation between the ETL and LTL stages, however, more studies need to be performed to test this idea.

Nerve activity and relative size:There are several areas that need to be addressed concerning description and discussion of nerve activity and size. One is strictly terminology of "nerve abundance" or "innervation" when in fact it is not really known what the innervation extent is within the limb – just the size of the host nerve. Thus the correlative finding is to relative contribution, or extent, of nerves provided to the regenerate – or within the regenerate as detailed in Figure 3. In Figure 3. it will be important to show absolute levels of nerve abundance (or normalized to body length), as well as the change, looks predominantly to be due to the addition of non-neural tissue leading change in of cross-sectional volume of the regenerate -- reduction of total mass of the early tiny limb compared to more mature limb will increase the fraction of nerve density but the absolute innervation is constant.

Reviewer 2 also pointed out concerns they had with the description of nerve activity and relative size. “One is strictly terminology of "nerve abundance" or "innervation" when in fact it is not really known what the innervation extent is within the limb – just the size of the host nerve. Thus, the correlative finding is to relative contribution, or extent, of nerves provided to the regenerate – or within the regenerate as detailed in Figure 3. In Figure 3. it will be important to show absolute levels of nerve abundance (or normalized to body length), as well as the change, looks predominantly to be due to the addition of non-neural tissue leading change in of cross-sectional volume of the regenerate -- reduction of total mass of the early tiny limb compared to more mature limb will increase the fraction of nerve density but the absolute innervation is constant.”

We agree with the point that the abundance of nerves directed toward the regenerating tissue does not necessarily correspond to the amount of innervation. We have altered the text related to these figures to make this distinction and will use “nerve density” when we refer to our direct measurements of nerves in the regenerating tissue. We have also removed “innervation” from the text and figures when we are not directly measuring nerve abundance. The data of absolute nerve abundance are provided in supplemental Figure 4. As Reviewer 2 suspected, the absolute nerve abundance is low in the ETL, and actually increases as the limb grows in size during regeneration. So, while the relative abundance of innervation decreases during these stages (because the limb grows), the number of nerves is increasing, indicating that the decrease in relative nerve abundance is due to the increase in tissue mass.

Is then growth regulation titering out of nerve-derived signals? This seems to be the argument, but is phrased as relative levels of nerve innervation or abundance – or even hyper-innervated, which seems to imply more nerves are entering or anastomosing in the limb in these stages. This has not been shown. If that is the specific statement the authors want to use, it begs the question if one raises innervation (added DRG graphs or extra nerve re-routing to a regenerative stump e.g. branchial plexus being diverted to the limb), can this extend growth of a regenerate through modulation of growth rate at the tiny limb phase??

Reviewer 2 asked, “Is then growth regulation titering out of nerve-derived signals? This seems to be the argument but is phrased as relative levels of nerve innervation or abundance – or even hyper-innervated, which seems to imply more nerves are entering or anastomosing in the limb in these stages. This has not been shown. If that is the specific statement the authors want to use, it begs the question if one raises innervation (added DRG graphs or extra nerve re-routing to a regenerative stump e.g. branchial plexus being diverted to the limb), can this extend growth of a regenerate through modulation of growth rate at the tiny limb phase?

Our current hypothesis is that the titering of nerve dependent signal(s) regulates the growth rate of the regenerate. We also agree that we have misdescribed the higher relative abundance of innervation in the ETL as “hyperinnervation.” We have removed reference to hyperinnervation from the text. Regarding Reviewer 2’s suggested experiment (grafting DRGs into an ETL staged limb to see whether growth was increased): we have performed that experiment and found that while the implanted DRG lives and innervates the regenerating tissue, the total abundance of nerves (grafted + endogenous) in the implanted limbs was the same as the non-grafted limbs. This made us suspect that there were additional layers of regulation at play regarding how nerve abundance is regulated within the host limb, which we felt was outside the scope of the current study. However, the ALM data from Figure 5—figure supplement 2, which showed the change in regenerating limb size over time on large and small hosts, shows that the change in limb size, particularly at the early time points (day 20-50), is greater on the large hosts. Although we did not directly measure the abundance of nerves in the regenerating limbs, this data indicates that the growth rate is positively correlated with nerve supply. We have further discussed this possibility in the Results section.

One more critical area is the description of nerve instructional information arising from nerve 'abundance' from Figure 5. This figure was hard for me to understand – in part due to titles of Panel 5c and 5d it was not clear what the difference between Graft Tissue Size and Donor Size is. But, beyond that the inference is that the larger animals have larger nerves at the stump (innervation abundance of the title) thereby being causative of the change. However, the findings are that host size determine size of the graft. The dependency of the nerve in this instruction is inferred by the requirement of the nerve for regenerative growth – but as the NM_ALM model with different sized axonal bundles shows (Figure 8), it is not instructional (at least when neural explants are used). It would be good if the authors could refine their language on abundance and innervation to be more precise on what is being measured in the experiment.

Reviewer 2 also said that “beyond that the inference is that the larger animals have larger nerves at the stump (innervation abundance of the title) thereby being causative of the change. However, the findings are that host size determine size of the graft. The dependency of the nerve in this instruction is inferred by the requirement of the nerve for regenerative growth – but as the NM-ALM model with different sized axonal bundles shows (Figure 8), it is not instructional (at least when neural explants are used). It would be good if the authors could refine their language on abundance and innervation to be more precise on what is being measured in the experiment.”

Our interpretation that nerves are instructing size are a combination of the observations from Figure 5 (ALM study) and Figure 7 (NM-ALM). The data from figure 5 shows that the size of the regenerate when grafted to ALMs on large hosts are larger than those grafted to small hosts. Although the size of the nerves routed to the regenerating limb is significantly larger on the large hosts (Figure 5—figure supplement 1), we did not rule out the possibility that there are other factors supplied by the host environment that could also impact growth. However, the experiment in Figure 7 shows that when the abundance of nerves supplying the regenerate remain constant, that the size of the host animal doesn’t impact the final size of the regenerate. Additionally, Figure 5—figure supplement 2 shows that the growth of the limbs on the NM-ALMs is not significantly different on the differently sized hosts. Together, these data indicates that non-neural extrinsic factors are not instructing size.

As Reviewer 1 pointed out, the data in Figure 8, which shows that NM-ALMs with different sized DRGs on the same sized host result in regenerates of the same size, indicates that nerve abundance alone does not instruct size. We think that there are three possible explanations for this observation. First, this indicates that the nerve is not autonomously regulating growth of the regenerate. It is possible that neural signaling is a conduit from some upstream signal, possibly from the CNS, that determines the rate of growth. In this situation, the regenerating tissue would not be directly responsive to that upstream signal, but rather the “interpreted” signal from the nerves. Alternately, it is possible that the tissue surrounding the nerves in their native environment are somehow required for the generation of growth regulating signals. Unfortunately, because we do not yet know the molecular nature of the nerve-dependent signal, we cannot measure whether the implanted DRGs expressed these at differing abundances at this time. The third possibility is that the motor neurons are required for the regulating allometric growth in the regenerate. The assays in figure 4 and 5 incorporated the modulation of signals from both sensory and motor neurons. However, the NM-ALM utilizes DRGs, which are composed of only the sensory neurons. The motor neurons were not included in this assay because the nerve bodies of these cells are located deep within the vertebrae and are technically very difficult to dissect for implantation without injuring the cells. Thus, it is also possible that the motor neurons instruct limb sizing. Our future studies will focus on the potential differential inputs from nerve types on the regulation of allometric growth. Additionally, we have discussed these possibilities in the text.

Patterns of Growth rate:Near the end of the paper, the authors present growth rate data over time for the regenerate of short limb and long limb groups. Figure S7 is really quite informative in demonstrating a shift in the rate of growth seen in Small hosts over that in Large hosts; the general rate of growth of regenerates from smaller hosts is initiated later -- this seems to be the crux of the difference. The result is that the limb never catches up to the size of limbs with small hosts but grows at generally the same rate. Could the authors integrate their findings with this latent initiation and nerve activity of the stump?

We were also interested in the observation that the rate of growth increases earlier in the large compared to small hosts. While there are multiple possible interpretations, it may be based on either the differential accumulation of a nerve-based signal in the grafted tissue, or differences in innervation rates. When the blastemas are harvested and grafted onto the ALM, they are likely to be without nerve signaling for at least a few days while the injured nerve regenerates and innervates the regenerating tissue. So, it is possible that the larger nerves take less time to either innervate or generate enough of the nerve-based signal to reinitiate growth in the regenerating tissue. We have added this interpretation into the text.